METHODS AND RESOURCES

# Atlas of the anatomical localization of atypical chemokine receptors in healthy mice

Serena Melgrati[1,2], Egle Radice[1], Rafet Ameti[1], Elin Hub[3], Sylvia Thelen[1], Pawel Pelczar[4], David Jarrossay[1], Antal Rot[3,5,6], Marcus Thelen[1]*

1 Institute for Research in Biomedicine, Università della Svizzera italiana, Bellinzona, Switzerland,
2 Graduate School for Cellular and Biomedical Sciences, University of Bern, Bern, Switzerland, 3 Centre for Microvascular Research, The William Harvey Research Institute, Queen Mary University London, London, United Kingdom, 4 University of Basel, Center for Transgenic Models, Basel, Switzerland, 5 Centre for Inflammation and Therapeutic Innovation, Queen Mary University London, London, United Kingdom, 6 Institute for Cardiovascular Prevention, Ludwig-Maximilians University, Munich, Germany

* marcus.thelen@irb.usi.ch

**Data Availability Statement:** All relevant data are within the paper and its Supporting Information files S1_Data (related to Fig 2C): https://doi.org/10.6084/m9.figshare.22362418 S2_Data (related to

## Abstract

Atypical chemokine receptors (ACKRs) scavenge chemokines and can contribute to gradient formation by binding, internalizing, and delivering chemokines for lysosomal degradation. ACKRs do not couple to G-proteins and fail to induce typical signaling induced by chemokine receptors. ACKR3, which binds and scavenges CXCL12 and CXCL11, is known to be expressed in vascular endothelium, where it has immediate access to circulating chemokines. ACKR4, which binds and scavenges CCL19, CCL20, CCL21, CCL22, and CCL25, has also been detected in lymphatic and blood vessels of secondary lymphoid organs, where it clears chemokines to facilitate cell migration. Recently, GPR182, a novel ACKR-like scavenger receptor, has been identified and partially deorphanized. Multiple studies point towards the potential coexpression of these 3 ACKRs, which all interact with homeostatic chemokines, in defined cellular microenvironments of several organs. However, an extensive map of ACKR3, ACKR4, and GPR182 expression in mice has been missing. In order to reliably detect ACKR expression and coexpression, in the absence of specific anti-ACKR antibodies, we generated fluorescent reporter mice, ACKR3^{GFP/+}, ACKR4^{GFP/+}, GPR182^{mCherry/+}, and engineered fluorescently labeled ACKR-selective chimeric chemokines for in vivo uptake. Our study on young healthy mice revealed unique and common expression patterns of ACKRs in primary and secondary lymphoid organs, small intestine, colon, liver, and kidney. Furthermore, using chimeric chemokines, we were able to detect distinct zonal expression and activity of ACKR4 and GPR182 in the liver, which suggests their cooperative relationship. This study provides a broad comparative view and a solid stepping stone for future functional explorations of ACKRs based on the microanatomical localization and distinct and cooperative roles of these powerful chemokine scavengers.

Fig 2D): https://doi.org/10.6084/m9.figshare.22362406 S3_Data (related to Fig 2F): https://doi.org/10.6084/m9.figshare.22362412 S4_Data (related to Fig 7A): https://doi.org/10.6084/m9.figshare.22362415 S5_Data (related to Fig 7F): https://doi.org/10.6084/m9.figshare.22362403 The flow cytometry data presented in the manuscript has been deposited at the FlowRepository (FR-FCM-Z65S).

**Funding:** The study was supported by grants from Swiss National Science Foundation (https://www.snf.ch Sinergia CRSII3_160719 ((MT, AR) and 310030_182727 (MT)), the Novartis Foundation for medical-biological Research (https://www.novartisfoundation.org/ #17B098 (MT)). Fondazione Fidinam, Lugano (https://www.fidinam.com/it/fondazione-fidinam MT) and the Helmut Horten Foundation (https://www.helmut-horten-stiftung.org/ MT) also provided support. Wellcome Trust Investigator Award 200817/Z/16/Z (A.R.) and Versus Arthritis Endowment (AR). The funders had no role in study design, data collection and analysis, decision to publish, or preparation of the manuscript.

**Competing interests:** The authors have declared that no competing interests exist.

**Abbreviations:** ACKR, atypical chemokine receptor; BM, bone marrow; BSA, bovine serum albumin; CAR, CXCL12-abundant reticulocyte; DARC, Duffy Antigen Receptor for Chemokines; eGFP, enhanced green fluorescent protein; FRC, fibroblastic reticular cell; GCB, germinal center B cell; GFP, green fluorescent protein; GPCR, G-protein–coupled receptor; KO, knockout; LEC, lymphatic endothelial cell; LN, lymph node; MFI, mean fluorescence intensity; MS, marginal sinus; MZ, marginal zone; MZB, marginal zone B cell; PCR, polymerase chain reaction; OCT, optimum cutting temperature; RFP, red fluorescent protein; RT-qPCR, reverse transcription quantitative PCR; SCS, subcapsular sinus; SMA, smooth muscle actin.

## Introduction

During inflammation and homeostasis, leukocyte trafficking is chiefly orchestrated by the chemokine system, along with their cognate and atypical receptors [1]. Chemokines are small (8 to 10 kDa) chemotactic cytokines that guide migrating cells during homeostasis and inflammation. Chemokines are classified into 4 families, based on the position of the first 2 of 4 conserved cysteines, which form disulfide bonds: CC, CXC, $CX_3C$, and XC [2]. Chemokines can promiscuously bind multiple canonical receptors, which trigger cell migration, or atypical chemokine receptors (ACKRs), which mainly act as scavengers.

ACKRs contribute to cell migration by binding and internalizing chemokines, delivering them to lysosomal degradation. This orchestrates the formation of locally confined gradients, which are necessary for efficient cell trafficking [3]. ACKRs are seven-transmembrane domain molecules that are structurally and phylogenetically related to G-protein–coupled receptors (GPCRs) but upon ligand binding are unable to signal though G-proteins, unlike conventional chemokine receptors [4]. Given the lack of G-protein activation, ACKR3 was proposed to have a signaling bias towards β-arrestins [5]. However, recent studies have challenged this hypothesis showing that β-arrestins are dispensable for ACKR3 internalization [6,7,8–10]. Four human and mouse ACKRs are currently known: ACKR1 through ACKR4. ACKR1, formerly known as Duffy Antigen Receptor for Chemokines (DARC), binds several inflammatory CC and CXC chemokines [2,11]. It is the most different and phylogenetically distant to the other ACKRs and canonical chemokine receptors [12]. ACKR1 expression was observed notably in erythrocytes, where it acts as a chemokine sink and as a reservoir and in nucleated erythroid cells where it regulates hematopoiesis [13]. In mouse venular endothelium, ACKR1 internalizes chemokines from the basolateral side and through transcytosis presents them at the luminal side [14]. ACKR2, also known as D6 or CKBP2, is also able to bind numerous inflammatory chemokines of the CC family [2,15]. More recently it was shown that ACKR2 can also bind the CXC chemokine CXCL10 [16]. Unlike ACKR1, ACKR2 is a scavenger receptor, which internalizes chemokines and delivers them for lysosomal degradation [17]. ACKR2 expression has been previously reported in the lung, skin, placenta syncytiotrophoblasts, lymphatic endothelium, and innate-like B cells [18–21].

ACKR3, previously named CXCR7, is mainly responsible for scavenging CXCL12 and CXCL11, resulting in their degradation [22–24]. ACKR3 expression was reported in a variety of cells and tissues including neurons, hematopoietic cells, platelets, umbilical vein endothelium and vascular endothelium, T cells, memory B cells, plasma blasts, and marginal zone (MZ) B cells [23,25–29]. The importance of ACKR3 is mainly attributed to its ability to scavenge CXCL12, and indeed, ACKR3 blockade increases CXCL12 levels systemically [30,31]. This scavenger receptor is necessary for survival, as global knockout (KO) in mice is perinatal lethal, due to a stenotic cardiac valve phenotype [32]. Moreover, it was demonstrated to be required for proper migration of primordial germ cells in zebrafish [33,34]. Tight regulation of CXCL12 availability is critical to maintain homeostasis in leukocyte and stem/progenitor cell retention and egress from the bone marrow (BM) and recruitment to sites of inflammation. Vascular endothelial expression of ACKR3 has been reported, allowing its immediate access to circulating CXCL12 in the bloodstream and suggesting that it is indeed acting as a scavenger for CXCL12 [31].

Finally, ACKR4, previously known as CCRL1, is able to bind and scavenge the chemokines CCL19, CCL20, CCL21, CCL22, and CCL25 [35–37]. It has been detected in cortical thymic epithelium, in keratinocytes, in the lung, and in lymphatic endothelial cells (LECs) [38–40]. Expression on the ceiling of the subcapsular sinus (SCS) LECs (cLECs) has been demonstrated to be essential for shaping CCL21 gradients and, consequently, lymph node (LN) homing of

dendritic cell as well as the reentry of B cells from the SCS back into the germinal centers [41–43]. ACKR4 has also been found to be critical for T cell migration in inflamed afferent lymphatics [44]. It was recently identified in mouse spleens in a three-dimensional sinusoidal network connected to the marginal sinus (MS), named the "peri-marginal sinus," which tightly surrounds the MZ [45]. ACKR4 mRNA expression was reported in germinal center B cells (GCB) and genetic ablation of the receptor in mice was associated with a hyperactivated B cell phenotype in ACKR$^{-/-}$ plasma blasts and GCBs [39,46], a finding that was not observed with another ACKR4-deficient mouse strain (ACKR$^{GFP/GFP}$) [38,47].

There are no selective chemokines for ACKRs as they share their ligands with one or more conventional chemokine receptor. To demonstrate ACKR activity in the presence of the respective conventional receptors and to overcome problems with the paucity of specific antibodies, we engineered fluorescent chimeric chemokines, namely CXCL11_12, which selectively binds ACKR3 [48], and CCL25_19, which selectively binds ACKR4 [36]. The chimeras are composed of the N-terminus of CXCL11 or CCL25 and the body of CXCL12 or CCL19, respectively, and contain at the C-terminus a tag for site-specific enzymatic labelling. The chimeras were used to demonstrate scavenging by the ACKRs in vivo.

Recently, a novel potential ACKR was identified. GPR182 is closely related to ACKR3 by phylogeny and was initially thought to be the receptor for adrenomedullin, a notion that has subsequently been refuted [49,50]. Two studies have demonstrated both in vitro and in vivo that GPR182 is able to bind and scavenge the chemokines CXCL9, CXCL10, CXCL12, and CXCL13 and could potentially interact with others [51,52]. GPR182 has been found to be preferentially expressed in vascular ECs, liver sinusoidal ECs, LECs, and intestinal stem cells and is up-regulated in tumor-associated LECs [53–55]. It has been found to act as an ACKR to prevent hematopoietic stem cell egress from the BM [51], and as a chemokine scavenger in the tumor microenvironment to limit T cell infiltration [52], and also as a negative regulator of hematopoiesis in zebrafish and mice, by regulating the leukotriene B4 biosynthetic pathway [56]. The recent discovery of GPR182 function as a chemokine scavenger that appears not to signal through G-proteins (51;52) should warrant it being renamed ACKR5.

Previous studies have shown that ACKR3, ACKR4, and GPR182 potentially share expression in endothelial cells. However, a wide-ranging map of active expression on structures where these receptors could truly act as scavengers have been missing, in large part due to the absence of specific validated functional antibodies. Through the use of double ACKR-expression reporter mice, ACKR-specific chimeric chemokines and a novel specific ligand for GPR182, we have generated an atlas of expression and coexpression of ACKR3, ACKR4, and GPR182 in healthy young mice. We focused primarily on endothelial cells, where these ACKRs can bind, internalize, and degrade readily available circulating chemokines to maintain tissue homeostasis. Noteworthy, these receptors share the ability to scavenge homeostatic chemokines.

## 2. Methods

### 2.1. Cells

Mouse 300.19 pre-B cells were cultured in B cell medium containing RPMI-1640 supplemented with 10% FBS, 1% PenStrep, 1% nonessential amino acids, 1% Glutamax, and 50 μM β-mercapto ethanol (β-ME). All cell culture media and supplements were from Gibco/Thermo Fisher. Cells stably expressing ACKRs or GPR182 were transfected using Amaxa Nucleofector (Lonza), along with a T2A-GFP sequence. Receptors were expressed fused via a self-cleaving peptide to green fluorescent protein (GFP), which splits posttranslationally to produce the 2 proteins and mark cells expressing the receptor.

## 2.2. Chemokine expression and purification

Recombinant chimeric chemokines were expressed, purified, and fluorescently labelled as previously described [36,48,57].

## 2.3. Binding and uptake assays by flow cytometry

All binding and uptake experiments were performed at 4°C or 37°C, respectively, for 45 minutes. Mean fluorescence intensity (MFI) was measured by flow cytometry. MFI was normalized to parental cells.

For binding and uptake assays, 300.19 pre-B cells stably transfected with receptors (and GFP in a T2A system), or parental, were incubated with 300 nM chimeric chemokines (all labeled in AF647 or Dy649) for 45 minutes. Cells were washed and analyzed by FACS (Fortessa, BD). The MFI of GFP, which is proportional to the level of receptor expressed [58], was used to normalize.

Displacement binding was measured by FACS by incubating 300.19 cells expressing GPR182-GFP with a fixed amount (5 nM) of fluorescently labeled (chimeric) chemokine and increasing concentrations of unlabeled (chimeric) chemokine. After 45 minutes, cells were washed and analyzed by FACS (Canto I, BD).

## 2.4. Mice

ACKR3$^{GFP/+}$ (https://www.jax.org/strain/008591) were obtained from Robyn Klein, and ACKR4$^{GFP/+}$ animals were obtained from Cornelia Halin [38]. ACKR3$^{GFP/+}$CD19$^{Cre/+}$ RFP STOP$^{fl/+}$ mice were generated as previously described [29].

GPR182$^{mCherry/+}$ mice were generated by CRISPR/Cas9 genome engineering in C57BL/6J mouse embryos, with mCherry knock-in replacing 1 GPR182 allele, resulting in a heterozygous reporter animal (S1 Methods, S1 Fig) [59,60]. By crossing 2 heterozygous reporter mice, we obtained GPR182-KO offspring (in mendelian ratio), in which both alleles of GPR182 were replaced with mCherry in homozygosity. Animals were crossed with ACKR3$^{GFP/+}$ and ACKR4$^{GFP/+}$ to obtain double reporter mice.

All animals were housed in specific pathogen-free facility, and all experiments were performed in accordance with the Swiss Federal Veterinary Office guidelines and authorized by the Animal Studies Committee of Cantonal Veterinary (Cantonal Committee for Animal Experimentation (CCEA, License: TI-33/2020), and under the license number PA672E0EE issued to QMUL by the UK Home Office. In addition, ACKR4$^{GFP/+}$ reporter mice were housed in specific pathogen-free facilities at Queen Mary University London, Charterhouse Square, or University of Birmingham, Edgbaston, UK. Experiments were performed using the tissues of 8- to 12-week-old mice, as approved by the Institutional Ethics and Animal Welfare Committees and the Home Office, UK. All mice used were in the C57BL/6 background. For the experiments, animals of both genders were used with no sex-dependent differences observed.

## 2.5. Immunofluorescence

Mice were killed in $CO_2$ and perfused first with 30 ml PBS, then with 10 ml PBS containing 2% PFA (MerckMillipore). Organs were harvested and fixed for 4 to 16 hours in 4% PFA at 4°C on a shaking platform. Organs were washed in PBS and embedded in 3% low-gelling temperature agarose (Sigma, Cat: A9414). Tissues were sectioned using a Leica Vibratome to produce 50 to 150 µm slices. Sections were blocked in Blocking Buffer (PBS, 1% FBS, 0.1% Tx-100, 0.01% NaN$_3$) for 1 hour at RT and then stained with selected primary or directly conjugated antibodies overnight in Blocking Buffer. Following two 20-minute washes, sections were

stained using secondary antibodies (1 hour RT) or DAPI (15 minutes RT), washed twice, and mounted on glass slides in FluoroMount (Sigma, Cat: F4680).

Cross-sections of colon and small intestines were removed, cleaned, and fixed in 2% PFA/PBS for 4 hours followed by 10%, 15%, and 30% sucrose/PBS for 3 hours or overnight. Tissues were subsequently frozen in optimum cutting temperature (OCT) compound on dry ice and stored at −80˚C. Around 5-μm thin sections were cut on a cryomicrotome, air dried, and then stored at −20˚C. Before staining, sections were rehydrated in 0.1% BSA/PBS and blocked with 10% goat serum for 30 minutes. Antibodies were diluted in 0.1% BSA/PBS and incubated on the sections for 40 minutes at RT. After 3 washes in 0.1% BSA/PBS, the sections were stained with DAPI (MilliporeSigma) for 5 minutes. After another washing step, the sections were mounted with Prolong Gold (Thermo Fisher Scientific) and cured overnight. Sections were imaged on a LSM800 (Zeiss).

For tissue clearing, agarose-embedded organs were sectioned using a Leica Vibratome to produce 500-μm slices. Sections were permeabilized and blocked in PBS containing 2% Tx-100, 10% serum, 0.05% $NaN_3$) for 24 hours at RT, then stained with antibodies over 48 hours at 4˚C. Sections were washed in wash buffer (PBS, 1% FBS, 0.1% Tx-100, 0.01% $NaN_3$) for 1 hour 3 times at RT. Sections were then cleared in RapiClear 1.52 (SUNJin Lab, Cat: #RC152001) until transparent following manufacturer's instructions. Alternatively, fixed spleens were washed in PBS and incubated overnight at 4˚C in a hydrogel solution (4% acryl-amide and 0.25% of 2,2′-Azobis[2-(2-imidazolin-2-yl)-propane]-dihydrochloride in PBS). The solution containing the spleens was saturated with nitrogen to remove oxygen and incubated at 37˚C for polymerization. Spleens were washed in PBS and cleared with the ACT-ECT method [61] (active clarity technique–electrophoretic tissue clearing containing 4% SDS and 200 mM boric acid (Logos)) for 5 hours. Spleens were washed overnight with PBS and incu-bated for 2 hours with cubic mounting media (50% sucrose, 25% urea, and 25% N,N,N′,N′-tet-rakis-(2-hydroxypropyl)-ethylenediamine) to increase transparency.

For BM slice preparation, femurs and tibiae were isolated, cleaned, and immersed in 2% PFA for 6 hours at 4˚C, then dehydrated in 30% sucrose for 72 hours at 4˚C, as previously described [62]. Bones were embedded in OCT and snap frozen in liquid nitrogen. Bones were sectioned longitudinally using a cryostat until the BM cavity was exposed. The OCT block con-taining the sample was reversed to repeat the procedure on the opposite side until the cavity was visible. Samples were incubated in blocking solution containing 0.2% Triton X-100, 1% fatty acid–free bovine serum albumin (BSA), 10% rabbit (Gibco, 16120–107) or goat serum (Gibco, 16210–064) in PBS overnight at 4˚C. Primary antibody staining were performed in blocking solutions for 3 days at 4˚C. Samples were then washed in PBS and incubated in block-ing solution containing secondary antibodies for 3 days. Following washing steps in PBS, sam-ples were then cleared in RapidClear for 6 hours. Samples or sections were imaged using either a Leica SP5 or a Stellaris 8 confocal microscope.

## 2.6. In vivo uptake

Mice were injected IV through the tail vein with a 150-μl solution containing 2.5-μM fluores-cent chemokine(s) in PBS. Animals were killed in $CO_2$ 30 minutes later, perfused, and organs harvested as described above.

## 2.7. Splenic endothelial cell isolation and RNA extraction

Freshly harvested spleens were injected with a digestion cocktail containing 0.1 mg/ml Dispase Grade I (Roche, Cat: D4818), 0.2 mg/ml Collagenase IV (Roche, Cat: 11088866001), 0.025 mg/ml DNAse I (Sigma-Aldrich, Cat: 10104159001) in RPMI 1640 (Gibco), and incubated at 37˚C

for 30 minutes. Spleens were transferred into fresh digestion mix and minced into small pieces, then incubated at 37°C for 20 minutes. The suspension was thoroughly mixed every 7 to 10 minutes. Digestion was quenched by adding cold Quench buffer (5 mM EDTA, 3% FBS in PBS). Cells were washed, and red blood cells lysed in ammonium-chloride-potassium lysis buffer. CD45$^+$ cells were removed using CD45 MicroBeads (Miltenyi Biotec, Cat: 130-052-301) and LS columns (Miltenyi Biotec, Cat: 130-042-401) following manufacturer's instructions. Cells were then stained with antibodies and analyzed by flow cytometry (Fortessa, BD).

RNA was extracted from sorted CD45$^-$ CD31$^+$ GPR182-mCherry$^+$ or GPR182-mCherry$^-$ or ACKR3-GFP$^{pos}$, ACKR3-GFP$^{mid}$, or ACKR3-GFP$^{neg}$ spleen cells using Quick-RNA Microprep kit (Zymo Research, Cat. R1050), following manufacturer's instructions. Briefly, pellets were lysed in RNA lysis buffer and purified in Zymo-Spin IC columns. DNAse was added to remove DNA contaminants. cDNA was reverse transcribed using qScript cDNA SuperMix (QuantaBio, Cat: 95048–025). RT-qPCR was performed using PerfeCTa SYBR Green FastMix (QuantaBio, cat: 95072–012), and ΔΔCT calculated using GAPDH as housekeeping gene.

### 2.8. Isolation of LECs, FRCs, and BECs from LNs

Freshly harvested LNs were isolated from mice and incubated in digestion mix containing RPMI, Liberase TL (Roche, Cat: 5401020001), DNAse I (Sigma-Aldrich, Cat: 10104159001) at 37°C for 1 hour with occasional shaking and thorough mixing. Single-cell suspensions were strained using a 70-μm cell strainer (Corning, Cat: CLS431751), centrifuged, and resuspended in αMEM (no nucleosides, + L-glutamine, Gibco, Cat: 12561056) supplemented with 10% FBS and 1% Penicillin/Streptomycin. Cells were then stained with antibodies and analyzed by flow cytometry (Fortessa, BD).

### 2.9. Isolation of bone marrow cells

BM was flushed from long bones with 5-ml RPMI containing 10% FBS. Single-cell suspensions were strained using a 70-μm cell strainer (Corning, Cat: CLS431751), centrifuged, and resuspended in ACK Lysing buffer according to the manufacturer's instructions (Gibco, Cat: A1049201) for red blood cell lysis. Cells were then stained with antibodies. For intracellular staining, cells were fixed and permeabilized using Cytofix/Cytoperm solution (BD Biosciences, Cat: 554722), then stained with antibodies. Cells were analyzed by flow cytometry (Fortessa, BD).

Antibodies used.

| Antibody target | Clone | Concentration | Manufacturer | Catalog number |
|---|---|---|---|---|
| Anti-CD31 | 390 | 1:200 | BioLegend | 102409 |
| Anti-CD31 | MEC13.3 | 1:200 | BioLegend | 102509 |
| Anti-CD31 PE | MEC13.3 | 1:200 | BioLegend | 102507 |
| Anti-LYVE-1 | 2125 | 1:100 | R&D | AF2125 |
| Anti-PDPN | 8.1.1 | 1:100 | BioLegend | 127415 |
| Anti-PNAd | Meca-79 | 1:200 | BioLegend | 120802 |
| Anti-CD45 | 30-F11 | 1:200 | BioLegend | 103116 |
| Anti-αSMA AF647 | 1A4 | 1:200 | Invitrogen | 50-9760-82 |
| Anti-αSMA efluor 570 | 1A4 | 1:200 | Invitrogen | 41-9760-80 |
| Anti-CD45 | 30-F11 | 1:500 | BioLegend | 103125 |
| Anti-Ter119 | TER-119 | 1:500 | BioLegend | 116231 |
| Anti-CXCL12 | K15C | 1:100 | | |

(*Continued*)

| Antibody target | Clone | Concentration | Manufacturer | Catalog number |
|---|---|---|---|---|
| Anti-CD140b | APB5 | 1:50 | BioLegend | 136010 |
| Anti-Sca1 | D7 | 1:200 | BioLegend | 108133 |
| Anti-Endomucin | V.7C7 | 1:200 | Santa Cruz Biotechnology | Sc-65495 |
| Anti-CD21/35 | 7E9 | 1:200 | BioLegend | 123407 |
| Anti-CD73 | TY/23 | 1:200 | BD Biosciences | 550738 |
| Anti-GFP | Polyclonal | 1:200 | Abcam | Ab6556 |
| Anti-CD117 | 2B8 | 1:200 | BioLegend | 105818 |
| Anti-Vimentin AF647 | EPR3776 | 1:200 | Abcam | Ab194719 |
| BV605 Goat anti-rat IgG | Polyclonal | 1:500 | BioLegend | 406522 |
| FITC Rabbit anti-rat IgG | Polyclonal | 1:500 | Southern Biotech | 6135–02 |
| Goat anti-rabbit IgG AF488 | Polyclonal | 1:500 | Life Technologies | A11008 |
| Streptavidin Pacific Blue | Polyclonal | 1:200–1:500 | Thermo Fisher | S11222 |

## Results

### Expression of ACKR3, ACKR4, and GPR182 in the red pulp of the spleen

Using tissue clearing approaches and confocal imaging, we analyzed the expression of ACKR3-GFP/GPR182-mCherry and ACKR4-GFP/GPR182-mCherry in the spleens of the respective double-reporter mice. The expression of all 3 scavengers, ACKR3, ACKR4, and GPR182, was observed in endothelial cells lining red pulp sinusoids (Fig 1). These were characterized by expression of the endothelial markers CD31, ICAM-2, and PVLAP (MECA-32) [45] (S2A and S2B Fig). As previously reported [45], ACKR4 expression (green) was observed predominantly restricted to the sinusoids of the perimarginal sinus, an interconnected sinusoidal network, which tightly surrounds the MZ (Fig 1A). ACKR3 expression (green) was present also in the perimarginal sinus, but more broadly in the sinusoids at the periphery of the red pulp (Fig 1B). ACKR3 is also known to be expressed on CD19[+] B cells in the MZ [29] where its expression could be observed in the reporter mice (Fig 1B, left panels, indicated by gray arrows, S2C and S3A Figs). Instead ACKR4 was identified in FAS[+] GL7[+] germinal center B cells as previously reported (S3B Fig) (46;47). By contrast, GRR182 was not detected in the CD45[+] compartment in either spleen or LN of healthy mice (S3C Fig). Using tissue clearing methods on the spleens of the ACKR3-GFP reporter mouse, we were able to visualize an interconnected 3D network of ACKR3-expressing vessels, from sinusoids, through their confluence up to the large veins (Fig 1C and 1D). Clearing of spleens from ACKR3-GFP/GPR182-mCherry and ACKR4-GFP/GPR182-mCherry mice that were pulsed with anti-CD19 antibodies shortly before sacrifice to stain the outer blood-exposed B cells, mainly MZ B cells, of the follicles [29] confirmed the colocalization of the ACKRs in 3D.

The GPR182-expressing sinusoids (red) were found to be most widely distributed, extending from the perimarginal sinus and throughout the whole red pulp (Fig 1A, 1B and 1D, right panels). Moreover, we noticed that while ACKR3 was only expressed in larger sinusoids (Fig 1B and 1D, right panels, indicated by white arrows in Fig 1B), GPR182 expression was also observed in the smallest vessels (Fig 1B and 1D).

To more accurately confirm the coexpression between populations of GPR182[+] and ACKR3[+] or ACKR4[+] sinusoids, we isolated CD45[−] CD31[+] cells following enzymatic digestion of spleens from the double reporters GPR182-mCherry–ACKR3-GFP and GPR182-mCherry–ACKR4-GFP. Here, we found populations with varying degrees of receptor expression and coexpression. Notably, GPR182 was found in 80% to 90% of all CD31[+] cells in the spleens

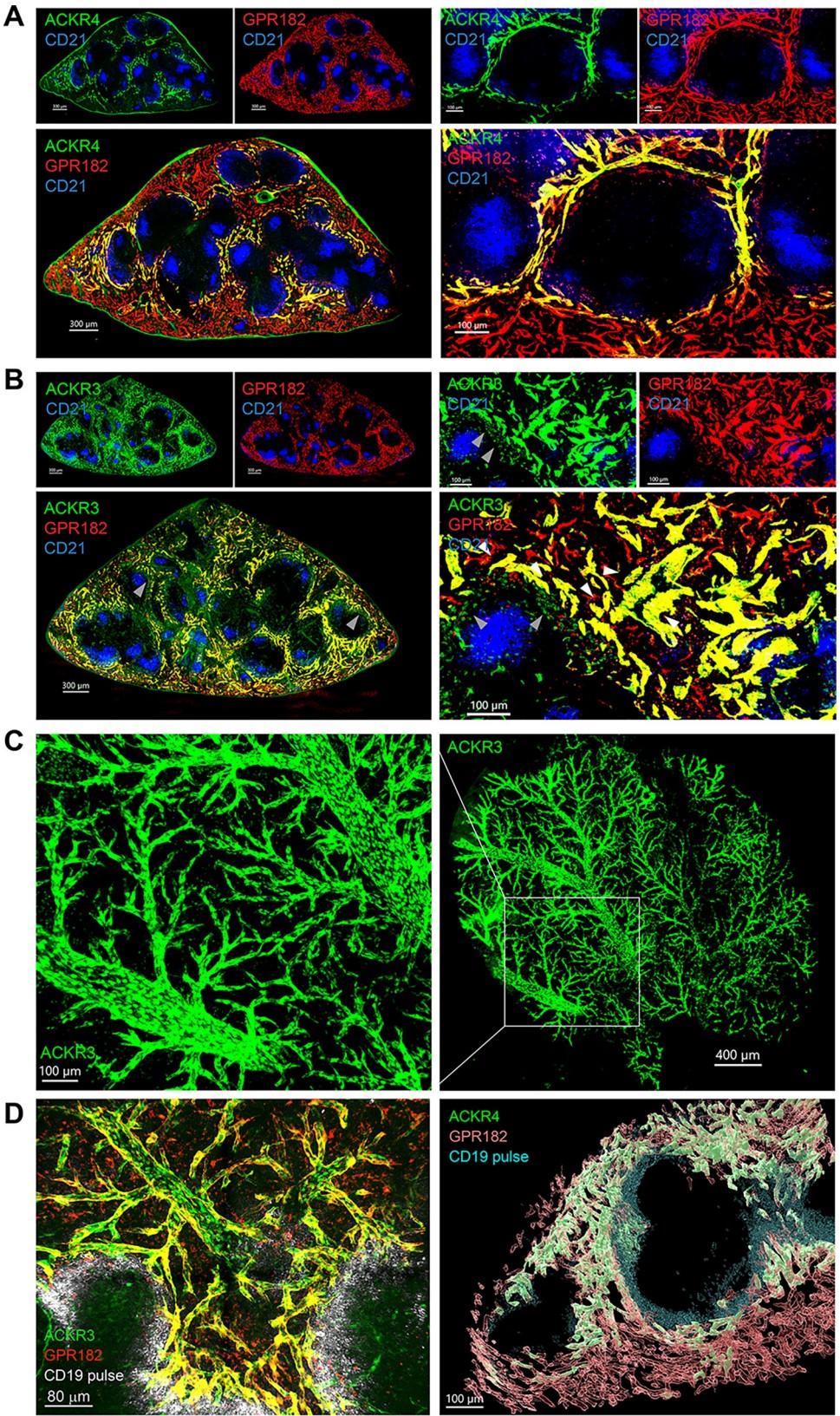

**Fig 1. ACKR3, ACKR4, and GPR182 are coexpressed in splenic sinusoids.** Confocal imaging of ACKR4$^{GFP/+}$ – GPR182$^{mCherry/+}$ and ACKR3$^{GFP/+}$ – GPR182$^{mCherry/+}$ in the red pulp of spleens. Blue staining mark (CD21/35) the

B cell follicles in the white pulp. (**A**) Expression of ACKR4 (GFP, green) in vessels of the perimarginal sinus and GPR182 (mCherry, red) more broadly in the sinusoidal network (left images scale bar = 300 μm). Sinusoids expressing both receptors appear in yellow in merged images. Right panels show enlargements to better visualize sinusoids (scale bar = 100 μm). (**B**) ACKR3 (GFP, green) is also found in the sinusoidal network, coexpressed with GPR182 (B, yellow–merge). On the left, the arrow points to MZ B cells expressing ACKR3 (scale bar = 300 μm). On the enlargement on the right, the arrows point to smaller GPR182-expressing sinusoids (scale bar = 100 μm). (**C**) Tissue clearing (ATC-ECT) of ACKR3$^{GFP/+}$ (GFP, green) spleen allows deep confocal imaging to reveal a branched network of ACKR3-expressing vessels on the right (scale bar = 400 μm), with enlargement on the left (scale bar = 100 μm). (**D**) Sinusoids of cleared spleens (RapiClear) from ACKR3$^{GFP/+}$ – GPR182$^{mCherry/+}$ (left) and ACKR4$^{GFP/+}$ – GPR182$^{mCherry/+}$ (right) were mosaic imaged over 200 μm (Z-axis) at 0.5-μm slice distance with 10× magnification. The borders of B cell follicles were marked with a pulse of anti-CD19 prior scarification (gray, left; cyan right). 3D reconstruction (right) was performed with Imaris software (scale bar = 80 μm, left, and 100 μm, right). GFP, green fluorescent protein; MZ, marginal zone.

(Fig 2A and 2B). With regard to ACKR3-coexpression (Fig 2A), we identified 4 populations: a GPR182$^{Hi}$ACKR3$^{Hi}$, comprising about 50% of all ECs, a GPR182$^{mid}$ACKR3$^{mid}$ (approximately 25%), and a smaller GPR182$^{mid}$ACKR3$^{Lo}$ population (approximately 10%), together with a limited (approximately 10%) double negative population. Conversely, coexpression of GPR182 and ACKR4 could be segregated into 3 main populations (Fig 2B): a double positive population (representing the ECs in the perimarginal sinus, approximately 37%), a larger GPR182$^+$ACKR4$^-$ population (approximately 48%) and a double negative population (13%). These results were confirmed by qPCR analysis of GPR182$^+$ ECs, which were analyzed for coexpression with ACKRs1 to 4 (Fig 2C). We also performed RNA-seq analysis of ACKR3-expressing EC populations (Fig 2D). Similarly, graded coexpression between ACKR3 and GPR182 was identified (Fig 2D, right panel), while ACKR3 and ACKR4 were coexpressed (albeit with lower copy numbers) in ACKR3$^{Hi}$ ECs and, to a lower extent, in ACKR3$^{mid}$ ECs (Fig 2D, third panel from left). While ACKR2, ACKR4, and GPR182 correlated with the expression of ACKR3, ACKR1 was only detectable in ACKR3 negative cells (Fig 2D, left panel).

By further analysis of endothelial cells that expressed GPR182, ACKR3, and ACKR4, we found that they were predominantly of venous nature, as they did not stain for smooth muscle actin (SMA, cyan, Fig 2E), a common marker of arteries and arterioles. Moreover, we performed qPCR analysis on the GPR182$^+$ CD31$^+$ cells, as these comprised the largest population (Fig 2F). Here, we found increased expression of the genes Ephrin B4 (Ephb4) [63], Neuropilin 2 (Nrp2) [64], and FMS-like tyrosine kinase 4 (Flt4) [65], which are widely considered markers of venous ECs. Instead, expression of the arterial markers Efnb2, Nrp1, Hey1, Hey2, and Notch4 [66–69] were insignificant. Taken together, these results support the view of a venous nature of GPR182$^+$ (and ACKR3$^+$ or ACKR4$^+$) endothelium in the spleen.

## Expression in the lymph node

We found the expression of ACKR3 in the LN to be minimal (Fig 3A), in both the CD45-positive (S3A Fig) and CD45-negative (S4A Fig) compartment. We could confirm ACKR4 expression in the LECs (S4B Fig), in particular in the ceiling of the SCS (Fig 3B), where it was shown to scavenge CCL21 to generate functional chemotactic gradients necessary for dendritic cell emigration from the sinus. Weak expression of GPR182-mCherry was also observed in the SCS, but in contrast to ACKR4, GPR182 was found in both ceiling and floor LECs (Fig 3B). This was confirmed by costaining with MAdCAM-1, which highlights LECs of the floor, or through imaging the SCS of the double reporter, in which ACKR4-GFP and GPR182-mCherry are colocalized in ceiling LECs (Fig 3B).

Interestingly, GPR182-mCherry was more broadly detected throughout the LN vasculature, in particular in CD31$^+$ cells of both blood and lymphatic nature (Figs 3A, 3B and S4

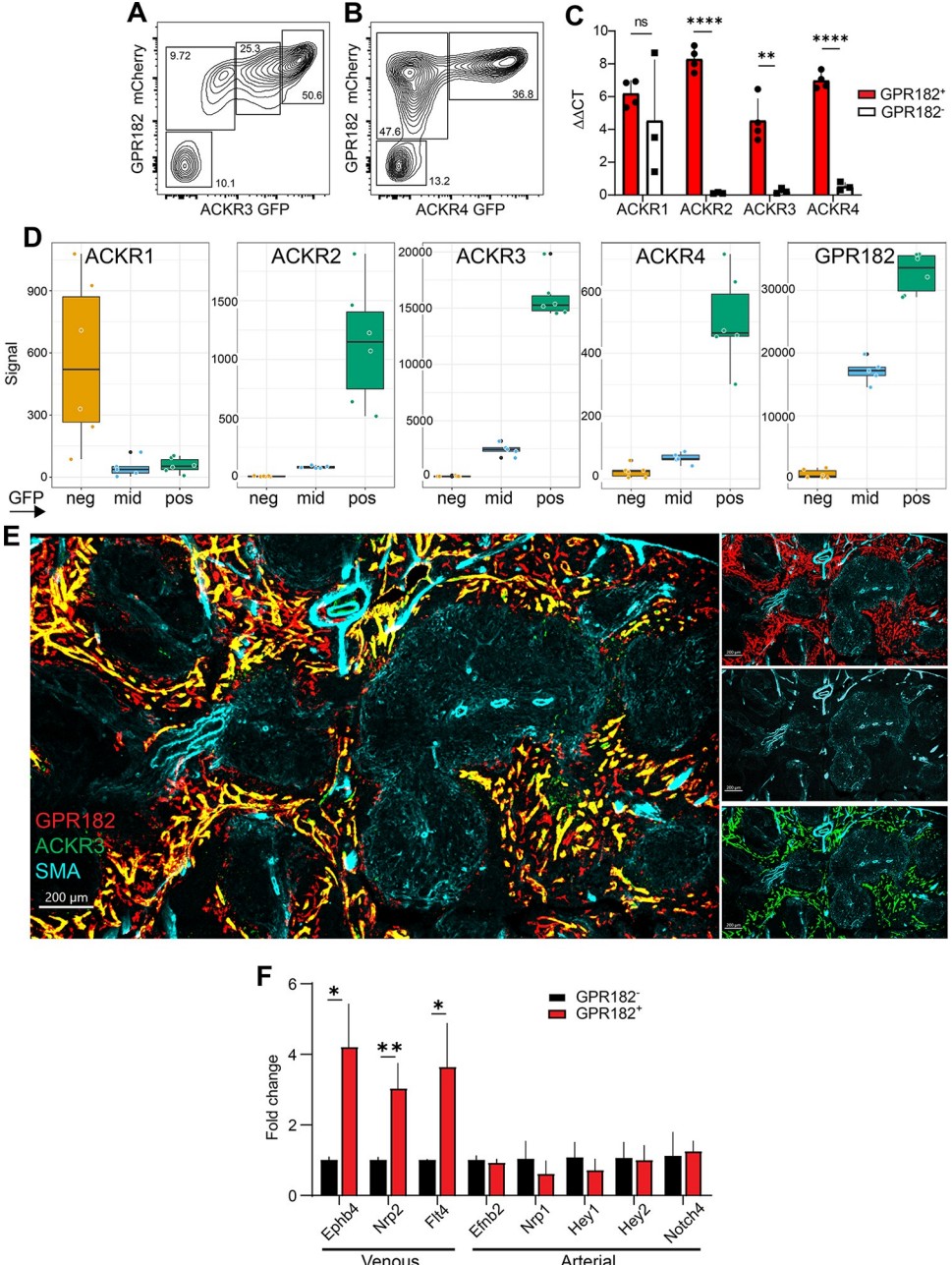

**Fig 2. GPR182⁺ splenic vessels are predominantly of venous nature.** Flow cytometry analysis of CD45 negative, CD31 positive splenic endothelial cells reveals coexpression of (**A**) ACKR3$^{GFP/+}$ – GPR182$^{mCherry/+}$ and (**B**) and ACKR4$^{GFP/+}$ – GPR182$^{mCherry/+}$. (**C**) RT-qPCR analysis of ACKR1–4 mRNA levels in GPR182⁺ and GPR182⁻ splenic endothelial cells (sorted CD45⁻ CD31⁺). ΔΔCT was calculated using GAPDH as housekeeping gene ($n = 3$, 3 pooled spleens per group. Error bars ± SD. Student $t$ test, $^{**}p < 0.01$, $^{****}p < 0.0001$). (**D**) ACKR1–4 and GPR182 (from left to right) RNA levels by RNA-sequencing in 3 ACKR3-expressing (GFP) endothelial cell populations (ACKR3 neg (ative), ACKR3 mid, and ACKR3 pos(itive)) of the spleen. Note the different scaling. ($n = 6$/group). (**E**) ACKR3$^{GFP/+}$ – GPR182$^{mCherry/+}$ splenic sinusoids do not colocalize with SMA staining (cyan) by confocal imaging (scale bar = 200 μm). (**F**) RT-qPCR analysis of venous and arterial markers in GPR182⁺ splenic endothelial cells relative to GPR182⁻ cells ($n = 3$, 3 pooled spleens per group. Error bars ± SD. Student $t$ test, $^{*}p < 0.05$, $^{**}p < 0.01$). FCS files and gating strategies are available in FlowRepository (Fig 2A and 2B). S1_Data (related to Fig 2C): https://doi.org/10.6084/m9.figshare.22362418; S2_Data (related to Fig 2D): https://doi.org/10.6084/m9.figshare.22362406; S3_Data (related to Fig 2F): https://doi.org/10.6084/m9.figshare.22362412. GFP, green fluorescent protein; RT-qPCR, quantitative reverse transcription PCR; SMA, smooth muscle actin.

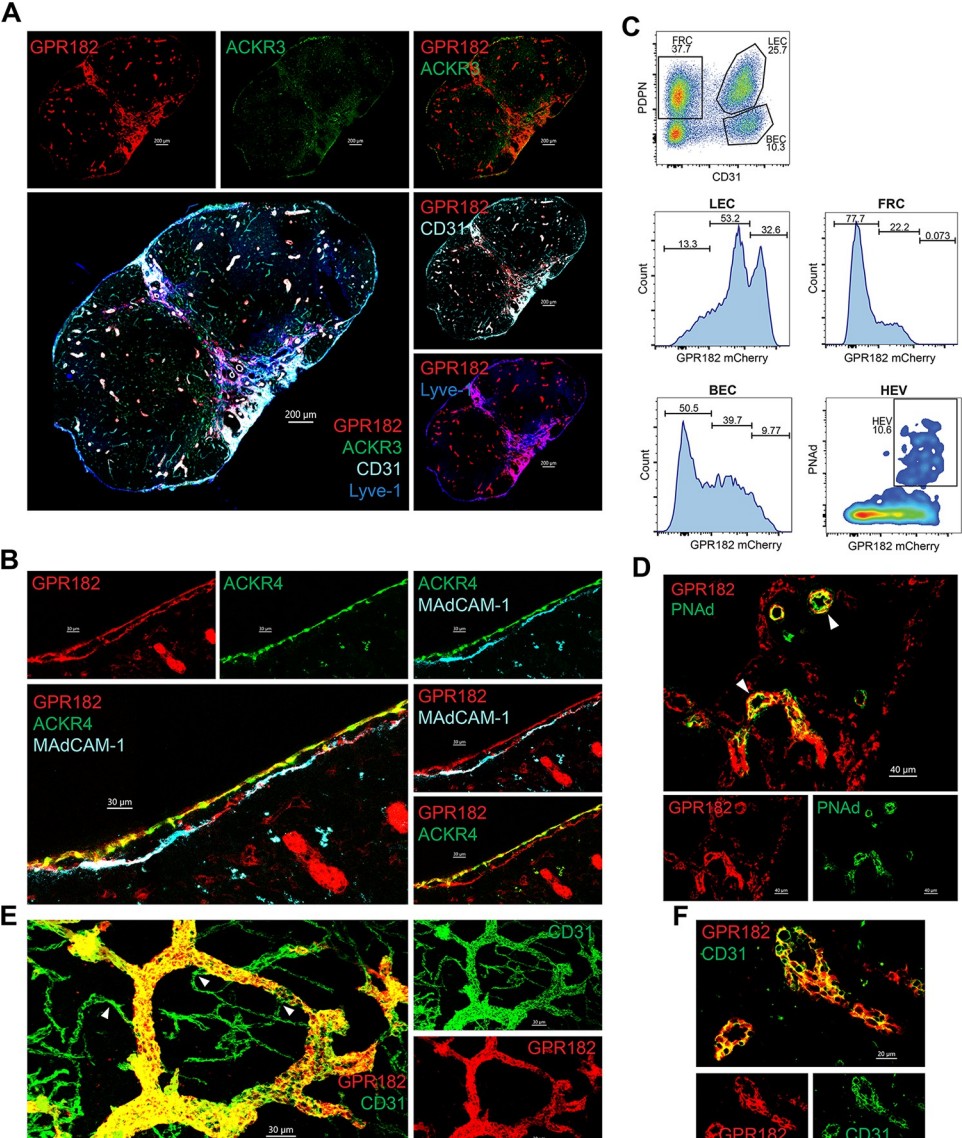

**Fig 3. Expression of chemokine scavengers in the LN.** (**A**) Confocal imaging of inguinal LN section of an ACKR3<sup>GFP/+</sup> – GPR182<sup>mCherry/+</sup> mouse, showing GPR182 in red (mCherry) and ACKR3 in green (GFP), counterstaining for CD31 (cyan) and Lyve-1 (blue) (scale bar = 200 μm). (**B**) Enlargement of confocal images of the SCS showing ACKR4<sup>GFP/+</sup> (GFP, green) and GPR182<sup>mCherry/+</sup> (mCherry, red) expression. Counterstaining of floor ECs with MAdCAM-1 (cyan) (scale bar = 30 μm). (**C**) Representative flow cytometry analysis of isolated LN stromal cells (CD45 negative) stained for CD31 and PDPN. Histograms show frequency of GPR182 cells gated as in top panel. HEV were gated on PDPN⁻ CD31⁺. (**D**) High magnification confocal images showing GPR182⁺ (mCherry, red) and PNAd⁺ (green) on HEV, arrows point to HEVs (scale bar = 40 μm). (**E**) Tissue clearing (RapiClear) of LN CD31⁺ capillaries (green, pointed to by arrows) and HEV (GPR182⁺, red/yellow merged) (scale bar = 30 μm). (**F**) Enlarged images show GPR182⁺ on HEV with typical cuboidal morphology (scale bar = 20 μm). FCS files and gating strategies are available in FlowRepository (Fig 3C). GFP, green fluorescent protein; LN, lymph node; SCS, subcapsular sinus.

C). By FACS analysis (Fig 3C) of digested tissues, we found high GPR182 expression in HEVs (identified by PNAd staining; Fig 3C and 3D, bottom panels) and lymphatic endothelium (Lyve-1⁺ PDPN⁺; Fig 3C, central left panels), which includes the aforementioned SCS. In contrast, fibroblastic reticular cells (FRCs; central right), identified as CD31⁻ PDPN⁺, were negative for GPR182. Nevertheless, confocal microscopy of cleared tissue revealed that not all

CD31$^+$ vessels express GPR182 (Fig 3E). More specifically, GPR182 was clearly identified in high endothelial cells, characterized by their unique cuboidal morphology (Fig 3F); however, the smaller interconnected CD31$^+$ capillaries (green, indicated by the white arrows (Fig 3E)) were predominantly negative for this receptor.

## Expression in primary lymphoid organs

We investigated the expression of the ACKR3, ACKR4, and GPR182 in the generative lymphoid organs, the BM and the thymus. In the thymus, GPR182 decorated the majority of the vasculature (CD31$^+$), while ACKR3 could be seen at low levels in very few GPR182-negative vessels (Fig 4A). More detailed analysis revealed that ACKR3 was present in both lymphatic

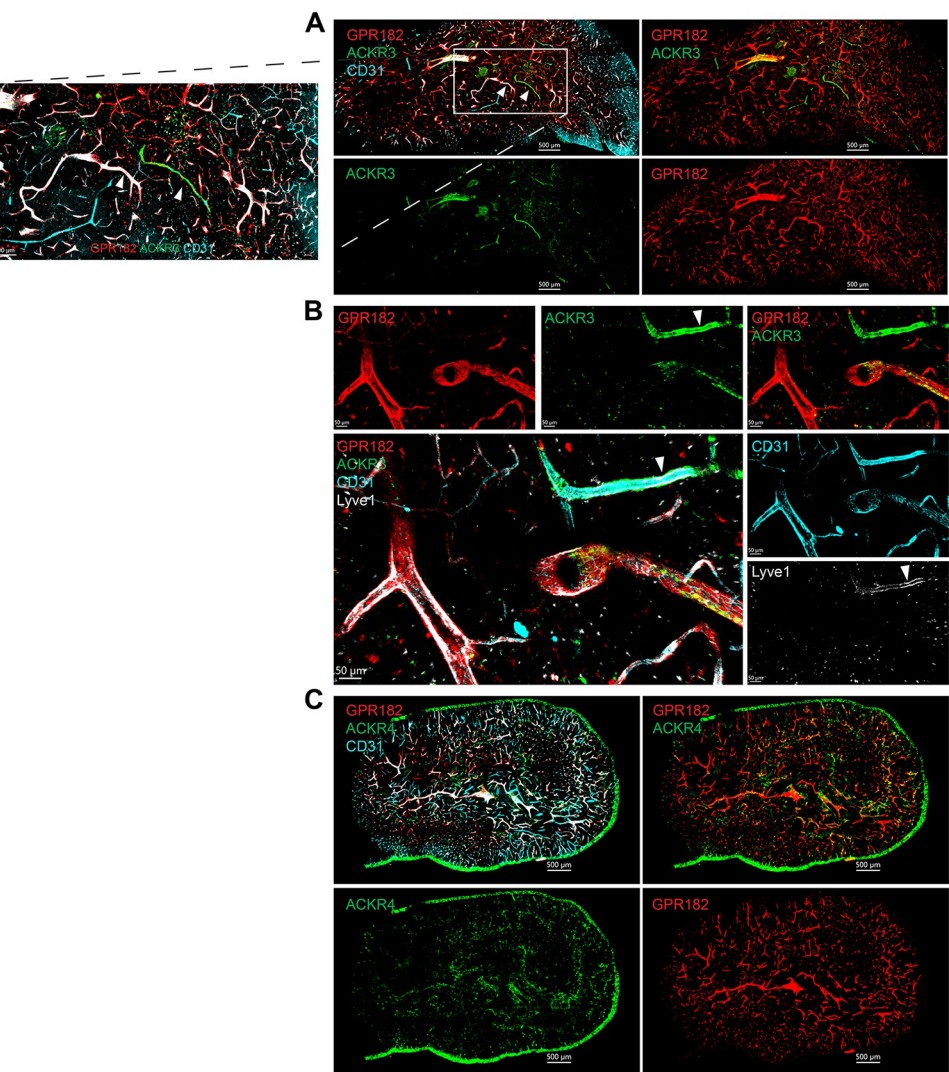

**Fig 4. GPR182 is expressed in the thymic vasculature, ACKR3 in lymphatics, and ACKR4 in thymic epithelium.** (**A**) Confocal imaging of a thymus section from an ACKR3$^{GFP/+}$ – GPR182$^{mCherry/+}$ mouse, showing GPR182 in red (mCherry) and ACKR3 in green (GFP) and CD31 in cyan (scale bar = 500 μm), with enlargement of the area in the white box on the left (scale bar = 200 μm). (**B**) High magnification of thymic vessels, showing colocalization of ACKR3 (GFP, green) with Lyve-1 (gray) and CD31 (cyan), pointed by the arrow (scale bar = 50 μm). (**C**) Imaging of a thymus section from an ACKR4$^{GFP/+}$ – GPR182$^{mCherry/+}$ mouse, showing ACKR4 (GFP, green) in the epithelium (scale bar = 500 μm). GFP, green fluorescent protein.

vessels (Lyve-1$^+$) and very few blood vessels (CD31$^+$ Lyve-1$^-$) (Fig 4A and 4B). In contrast, GPR182 was predominantly found in Lyve-1–negative blood vessels. ACKR4, conversely, was identified in the cortical and medullary thymic epithelial cells, as previously described (Fig 4C) [70].

We imaged the BM following tissue clearing. GPR182 was found in endomucin$^+$ CD31$^+$ sinusoids within the marrow (Fig 5A and 5B). In contrast, ACKR3 was found primarily in the bone (Fig 5A). Interestingly, we identified ACKR4 in CXCL12-abundant reticulocytes (CARs) (Fig 5C–5E). High-magnification imaging and 3D rendering showed that these were present in close association with GPR182$^+$ sinusoids (Fig 5D and 5E). We also identified the ACKR4 CXCL12-expressing (K15C$^+$) CAR cells by flow cytometry (CD45$^-$ Ter119$^-$ Sca1$^-$ CD31$^-$). The observations are in agreement with scRNA-seq-based findings by Baccin and colleagues, who identified ACKR3 in endosteal fibroblasts, arteriolar fibroblasts, and myofibroblasts; GPR182 in sinusoidal ECs; and ACKR4 in adipo-CAR and osteo-CAR (and at lower levels in Ng2$^+$ mesenchymal stem cells) [71].

## Expression in intestine and colon

Analysis of intestine and colon revealed common and distinct patterns of ACKR expression (Figs 6 and S5). In the small intestine, GPR182 was mainly expressed in the villi in lacteals (Lyve-1$^+$) and in the lymphatic vessels of the submucosa, and only marginally in CD31$^+$ capillaries in the villi (Figs 6A, 6B, S5A and S5B). ACKR3, instead, was found in CD31$^+$ endothelium of both the villi and the submucosa of the small intestine (Fig 6A). In contrast, ACKR4 was found in non-endothelial Lyve-1 and CD31-negative cells of the submucosa (Fig 6B). Closer inspection revealed that ACKR4$^+$ cells were present above, below, and within the outer longitudinal smooth muscle cell layer, as well as within the serosal layer (Fig 6C), in direct contact with interstitial c-kit$^+$ Cajal cells (Fig 6D) and colocalized with vimentin, suggesting their mesenchymal origin (S5E Fig). In the colon, ACKR3 was primarily found in the colonic vasculature (Figs 6E and S5C). Some coexpression with GPR182 was seen; however, the latter was mainly found in the submucosa, while ACKR3 was observed mostly in the vessels of the mucosa of the colon. Conversely, ACKR4 expression in the colon resembled its unique expression in the small intestine, namely in the submucosa (above, below, and within) and subserosa (Figs 6F, 6G and S5D). As in the small intestine, these cells were found in close proximity to Cajal cells and were in part vimentin positive (Figs 6H and S5E).

## Use of ACKR-semi-specific chimeras to reveal scavenging activity

To study the surface expression of the 3 receptors and their internalization capacity, we generated chimeric chemokines for each receptor; as due to the promiscuity of the chemokine system, specific ligands were missing. We had previously generated the chimeras CXCL11_12 and CCL25_19 for ACKR3 and ACKR4, respectively (36;48). The chimeras consist of the N-terminus of CXCL11 or CCL25 and the body of CXCL12 or CCL19. Both CXCL11 and CXCL12 also bind CXCR3 and CXCR4, respectively. CCL25 and CCL19 are specific ligands for CCR9 and CCR7, respectively. We therefore sought to design and produce a GPR182-specific chimeric chemokine. Considering that the spectrum of chemokines bound by GPR182 has been suggested to be generally broader than for other ACKRs, we selected the N-terminus of human CXCL11, composed of 8 amino acids, and the body of murine CCL20. We specifically chose to hybridize a chemokine of the human CXC family to one of the murine CC family to reflect the broad scavenging ability of this novel receptor. The chimeric chemokine was named CXCL11_20 (S6 Fig). Despite the marked differences in their primary amino acid sequences, the 3 chimeras shared biophysical properties, including the disulfide bridges,

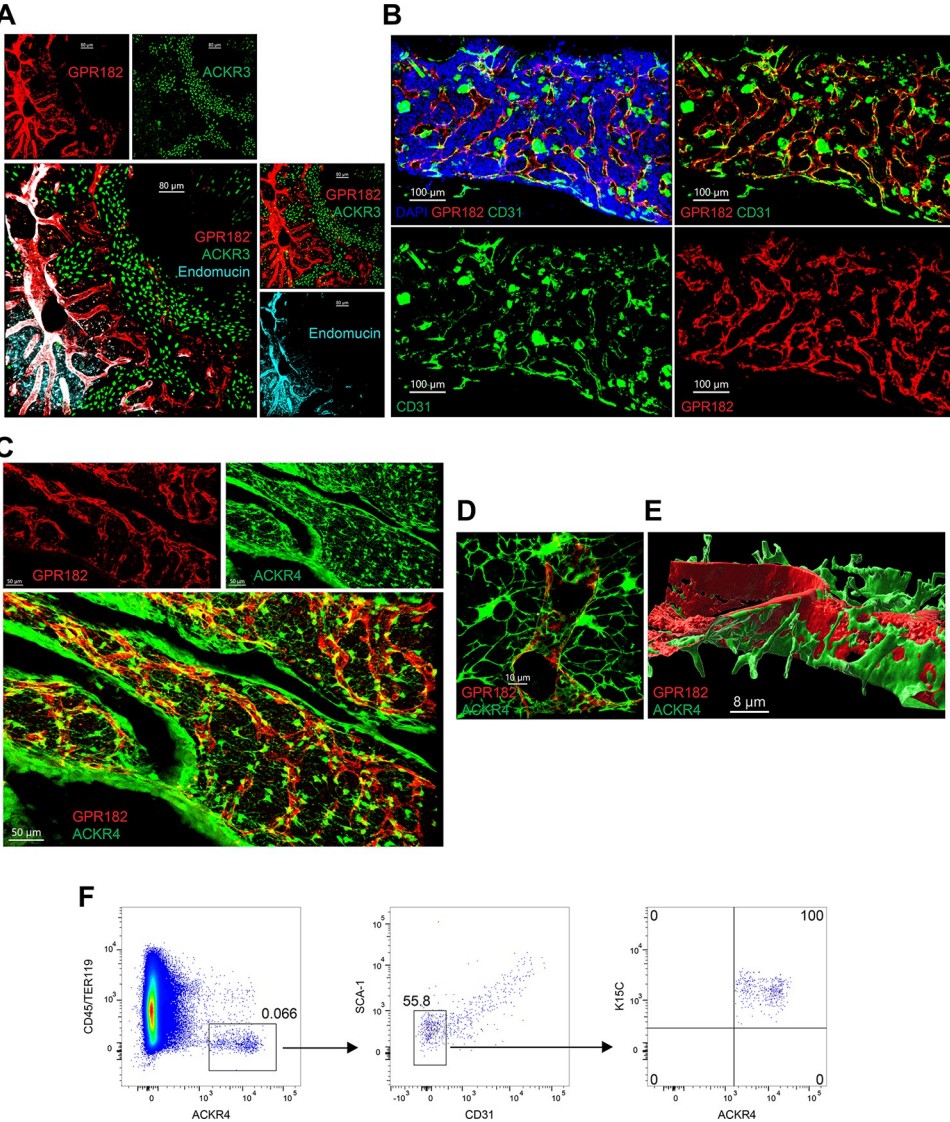

**Fig 5. Expression of chemokine scavengers in the BM. (A)** Confocal images of cleared (RapiClear) BM tissue, with GPR182 in red (mCherrry), ACKR3 in green (GFP), and Endomucin in cyan (scale bar = 80 μm). **(B)** Imaging of BM sinusoids expressing CD31 (green), GPR182 (mCherry, red), and DAPI (blue) (scale bar = 100 μm). **(C)** Imaging of BM (RapiClear) from a ACKR4^GFP/+ – GPR182^mCherry/+ mouse, with GPR182 in red (mCherry) and ACKR4 in green (GFP) (scale bar = 50 μm). **(D)** High magnification showing GPR182+ sinusoids (mCherry, red) surrounded by ACKR4+ CAR cells (green) (scale bar = 10 μm). **(E)** 3D reconstruction (Imaris software) of CAR cells surrounding sinusoid (scale bar = 8 μm). **(F)** Flow cytometry analysis of BM cells: ACKR4^GFP/+ cells are negative for CD45 and Ter119 (left), CD31 and Sca1 (center), but positive for CXCL12 (stained with K15C) (right). FCS files and gating strategies are available in FlowRepository (Fig 5F). BM, bone marrow; CAR, CXCL12-abundant reticulocyte; GFP, green fluorescent protein.

molecular weight range (9.6 to 13.2 kDa), isoelectric points (pI 9.5 to 10.5), and they eluted similarly between 32% and 40% acetonitrile from C18 reverse phase columns.

The specificity of CXCL11_20 for GPR182 was tested by flow cytometry (Fig 7A). We incubated CXCL11_20 labelled with Atto565 with mouse 300.19 pre-B cells transfected with ACKR1, ACKR2, ACKR3, ACKR4, and GPR182 (expressed using a T2A-GFP system) at 4°C to measure binding and at 37°C to measure uptake. At 4°C, plasma membranes are stiff and receptor endocytosis is blocked; therefore, only surface receptors are available for binding.

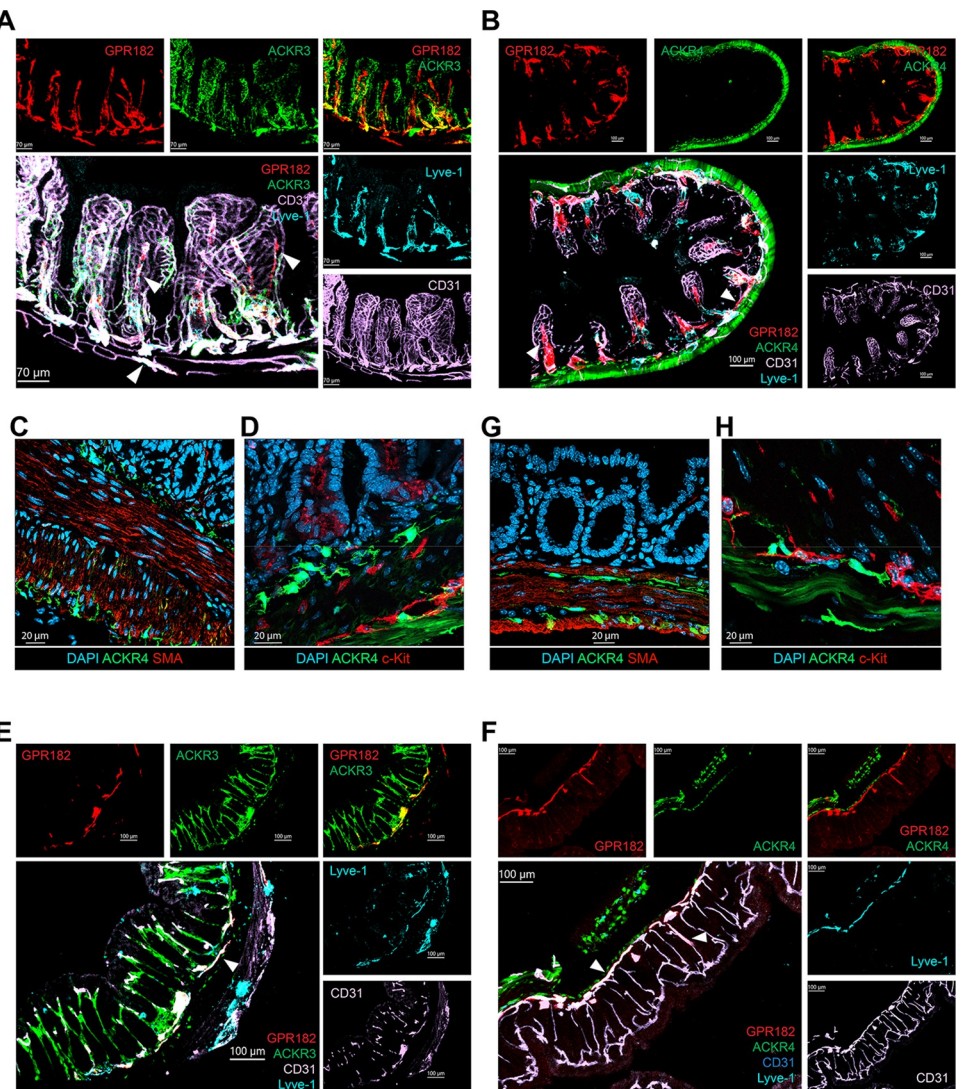

**Fig 6. Expression of chemokine scavengers in small intestine and colon.** Confocal imaging of small intestine from (**A**) ACKR3^{GFP/+} – GPR182^{mCherry/+} mouse, GPR182 (mCherry, red), ACKR3 (GFP, green), CD31 (pink), and Lyve-1 (cyan) (scale bar = 70 μm), and (**B**) ACKR4^{GFP/+} – GPR182^{mCherry/+} mouse, GPR182 (mCherry, red), ACKR4 (GFP, green), CD31 (pink), and Lyve-1 (cyan) (scale bar = 100 μm). Arrows point to central lacteal (GPR182+, mCherry, red) and submucosal lymphatics (left and right). (**C**) Immunofluorescence image of a section from small intestine from an ACKR4^{GFP/GFP} mouse, showing ACKR4 expressing cells (GFP, green). Smooth muscle cells are positive for α-SMA (red), and cell nuclei in DAPI (blue), scale bar: 20 μm. (**D**) Immunofluorescence image of a section from small intestine from an ACKR4^{GFP/GFP} mouse, showing ACKR4 expressing cells (GFP, green) in close proximity to c-kit–positive Cajal cells (red); cell nuclei DAPI (blue), scale bar: 20 μm. (**E**) Confocal images of colon from an ACKR3^{GFP/+} – GPR182^{mCherry/+} mouse, GPR182 (mCherry, red), ACKR3 (GFP, green), CD31 (pink), and Lyve-1 (cyan) (scale bar = 100 μm). (**F**) Confocal images of colon from an ACKR4^{GFP/+} – GPR182^{mCherry/+} mouse, GPR182 (mCherry, red), ACKR4 (GFP, green), CD31 (pink), and Lyve-1 (cyan); arrows indicate the lymphatic vessels (scale bar = 100 μm). (**G**) Immunofluorescence image of a section of colon of an ACKR4^{GFP/GFP} mouse, showing cells expressing ACKR4 (GFP, green) and α-SMA (red); cell nuclei DAPI (blue), scale bar: 20 μm. (**H**) Immunofluorescence image of a colon section from an ACKR4^{GFP/GFP} mouse, showing ACKR4 expressing cells (GFP, green) in close proximity to c-kit–positive Cajal cells (red); cell nuclei DAPI (blue), scale bar: 10 μm. GFP, green fluorescent protein.

GPR182 efficiently bound and took up the chimeric construct. Virtually, no binding and uptake was observed in cells transfected with ACKRs, except for marginal uptake from ACKR2.

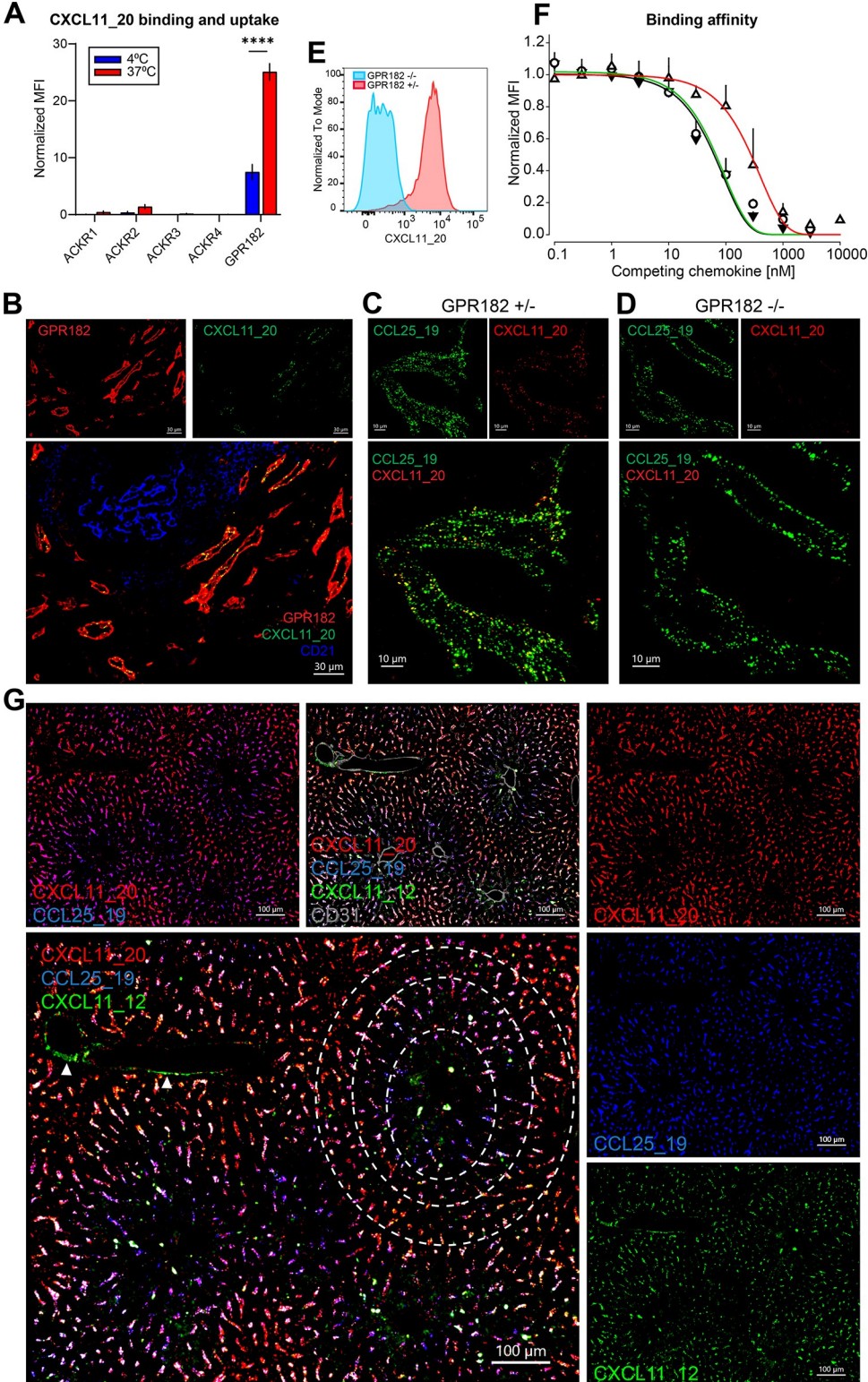

**Fig 7. Chimeric chemokines may be used to detect ACKR expression.** (**A**) In vitro binding and uptake assay: 300.19 pre-B cells expressing ACKRs or GPR182 were incubated with CXCL11_20 to test for binding (4°C, blue bars) or uptake (37°C, red bars). MFI was normalized to GFP expression ($n$ = 3, error bars ± SD). (**B**) Confocal images of uptake of IV-injected CXCL11_20 into endosomes (green) in spleen sinusoids from GPR182$^{mCherry/+}$ mouse (mCherry, red). Yellow shows colocalization in merged images, CD21/35 (blue) (scale bar = 30 µm). (**C**) Confocal

images of spleen sinusoids following coinjection of CCL25_19-Atto488 (green) and CXCL11_20-AF647 (red) GPR182$^{mCherry/+}$ heterozygous mice (scale bar = 10 μm). (**D**) Uptake of the same chemokines by GPR182-KO (GPR182$^{mCherry/mCherry}$) mice (scale bar = 10 μm). (**E**) CXCL11_20 in vivo binding and uptake by GPR182-proficient (red histogram) and GPR182-deficient (blue) spleen endothelial cells (gated on CD45$^-$ CD31$^+$). (**F**) Displacement binding of CXCL11_20 (black), CXCL11_12 (green), and CCL25_19 (red) on GPR182-expressing 300.19 pre-B cells ($n = 3 \pm$ SD). (**G**) Confocal image of a liver section from a wild-type mouse following triple coinjection of CXCL11_20 (red), CCL25_19 (blue), and CXCL11_12 (green). Purple shows colocalization between blue and red. Arrows point to central veins taking up CXCL11_12 via ACKR3. Dashed circles highlight areas of ACKR4 and GPR182 expression and coexpression (scale bar = 100 μm). FCS files and gating strategies are available in FlowRepository (Fig 7E). S4_Data (related to Fig 7A): https://doi.org/10.6084/m9.figshare.22362415; S5_Data (related to Fig 7F): https://doi.org/10.6084/m9.figshare.22362403. ACKR, atypical chemokine receptor; GFP, green fluorescent protein; MFI, mean fluorescence intensity.

The generation of chimeras allowed us to study the activity of each receptor in situ. To validate the new tool, we injected CXCL11_20 into a GPR182$^{mCherry/+}$ reporter mouse (Fig 7B). We isolated, fixed, and processed spleens for confocal imaging, 30 minutes later. We observed exact colocalization of endosome-like structures containing the chimera in cells expressing GPR182-mCherry.

To further test the specificity of CXCL11_20 for GPR182 in vivo, we injected CXCL11_20 labeled with AF647, together with the chimera for ACKR4, CCL25_19-Atto488, as a control, in GPR182$^{mCherry/+}$ (heterozygous (GPR182$^{+/-}$); Fig 7C) and GPR182$^{mCherry/mCherry}$ (both alleles replaced by mCherry, therefore KO (GPR182$^{-/-}$); Fig 7D) mice. Spleens were isolated, fixed, and processed for imaging, 30 minutes postinjection. As expected, both chemokines were taken up in heterozygous animals. However, GPR182-KO animals failed to take up CXCL11_20 (Fig 7C and 7D), and only endosomes containing CCL25_19 were identified, which indicate uptake by ACKR4, which is also expressed in these splenic sinusoids. Alternatively, endothelial cells were isolated from spleens by enzymatic digestion, CD45$^+$ cells depleted, and analyzed by flow cytometry by gating on CD45$^-$ CD31$^+$ cells. The CXCL11_20 fluorescence could only be detected with cells derived from GPR182$^{mCherry/+}$ mice but not in GPR182$^{mCherry/mCherry}$ (Fig 7E). Taken together, these results establish CXCL11_20 as a GPR182-specific chemokine.

The recent finding that GPR182 can broadly scavenge chemokines prompted us to test the ability of GPR182 to bind and scavenge CXCL11_12 and CCL25_19, which we had previously produced for ACKR3 and ACKR4, respectively. We performed competition binding experiments by incubating transfected 300.19 pre-B cells with a low concentration (5 nM) of fluorescent chimeric chemokine, which was outcompeted with increasing concentrations of its unlabeled version. GPR182 bound CXCL11_20 and CXCL11_12 with a calculated Kd of around 60 nM and 65 nM, respectively (Fig 7F). However, the affinity of CXCL11_12 for GPR182 was about 60-fold lower than the reported Kd for ACKR3 [48]. In case of CCL25_19, the affinity for GRP182 was even lower with a Kd around 300 nM, which is about 300 times less than the affinity of the chimera for ACKR4 [36].

## Chimera coinjection reveals polarized expression of GPR182 and ACKR4 in the liver

Imaging organs such as the liver using fluorescent reporter animals can pose a challenge given by the endogenous autofluorescence, especially in the GFP channel. Therefore, our chimeric ACKR ligands became useful to overcome this.

Taking advantage of the fact that the affinity of GPR182 for CCL25_19, the chimera designed for ACKR4, and CXCL11_12, the chimera designed for ACKR3, was relatively low, and that CXCL11_20 is specific for GPR182, we intravenously coinjected the 3 chimeras

labeled in different colors (Atto488, Atto565, or AF647) into a C57BL/6 mouse. After 30 minutes, livers were perfused and processed. As expected, the chimeras were found in seemingly endosomal structures within sinusoidal endothelial cells (Fig 7G). Interestingly, we noticed an apparently inverse gradient of uptake of either CCL25_19 or CXCL11_20, suggesting an inverse relationship of expression between ACKR4 and GPR182. More specifically, as shown within the white dashed circles, the sinusoids immediately surrounding central veins primarily displayed uptake of CCL25_19 (Fig 7G, in blue). In more distant sinusoids, we could find primarily "magenta" endosomes, which suggest concomitant uptake of CCL25_19 and CXCL11_20 (Fig 7G, in red). Further away, we observed mainly red endosomes, indicative of CXCL11_20 uptake. In other words, this suggests that ACKR4 is expressed in the sinusoids closest to central veins; ACKR4 and GPR182 are coexpressed (and are similarly active) in sinusoids that are found slightly further away; and GPR182 alone decorates the outermost sinusoids. The similar biophysical properties of the chimeras suggest a similar diffusion, further supporting different expression of the receptors in blood vessels. Inspecting ACKR4-eGFP expression in liver revealed weak signal and considerable autofluorescence background (S7A Fig); therefore, the signal was amplified using an anti-eGFP antibody (S7B Fig). The pattern of ACKR4 liver expression recapitulated the observations of CCL25_19 binding, with higher expression near central veins (S7B Fig) and diminishing towards the edges of lobules (S7B Fig). CXCL11_12 (Fig 7G, green) was more evenly dispersed but was found in endosomes in central veins (white arrows). Taken together, these data highlight that chimeric chemokines can be a valuable tool to study levels of active ACKR expression and their function in health and disease.

## Injection of chimeric chemokines reveals expression of ACKRs in the kidney

We combined the use of the ACKR reporter mice and the chimeric chemokines that we produced to study expression of ACKR3, ACKR4, and GRP182 in the kidney. By using double reporter ACKR3-GFP–GPR182-mCherry, we were able to identify expression of GPR182 in primarily in glomeruli and in the vascular pole of the Bowman's capsule (Fig 8A). Previous reports [54] identified GPR182 in podocytes. However, we found that GPR182 signal did not colocalize with podoplanin (PDPN) staining (Fig 8B), but rather with CD73 (Fig 8C), suggesting that GPR182 is expressed in both intraglomerular and extraglomerular mesangial cells. ACKR3 was also identified in the vascular pole and subsets of vessels but not in the intraglomerular mesangium (Fig 8A). We injected the chimera CXCL11_20 to identify GPR182 activity in an ACKR3$^{GFP/+}$ reporter mouse. By focusing onto the medullary region, we found high ACKR3-GFP expression in renal tubuli and uptake of CXCL11_20 by lining CD31$^+$ cells (Fig 8D). Moreover, coinjection of CXCL11_20 with CCL25_19 further revealed active ACKR4 expression in the vascular pole (Fig 8E), which was confirmed with the double reporter mouse for ACKR4-GFP–GPR182-mCherry (Fig 8F). Taken together, these data suggest that fluorescently labelled chimeric chemokines can be used to reveal active scavenger receptor expression in a number of organs, overcoming issues including autofluorescence, which can prevent successful imaging when using a reporter mouse.

## Discussion

ACKRs are critical in maintaining homeostasis and generate tightly regulated local chemokine gradients through binding, internalizing, and delivering chemokines for lysosomal degradation. Here, we generated an extensive atlas of active ACKR expression for the 3 scavengers ACKR3, ACKR4, and GPR182. To study ACKR expression in the absence of functional

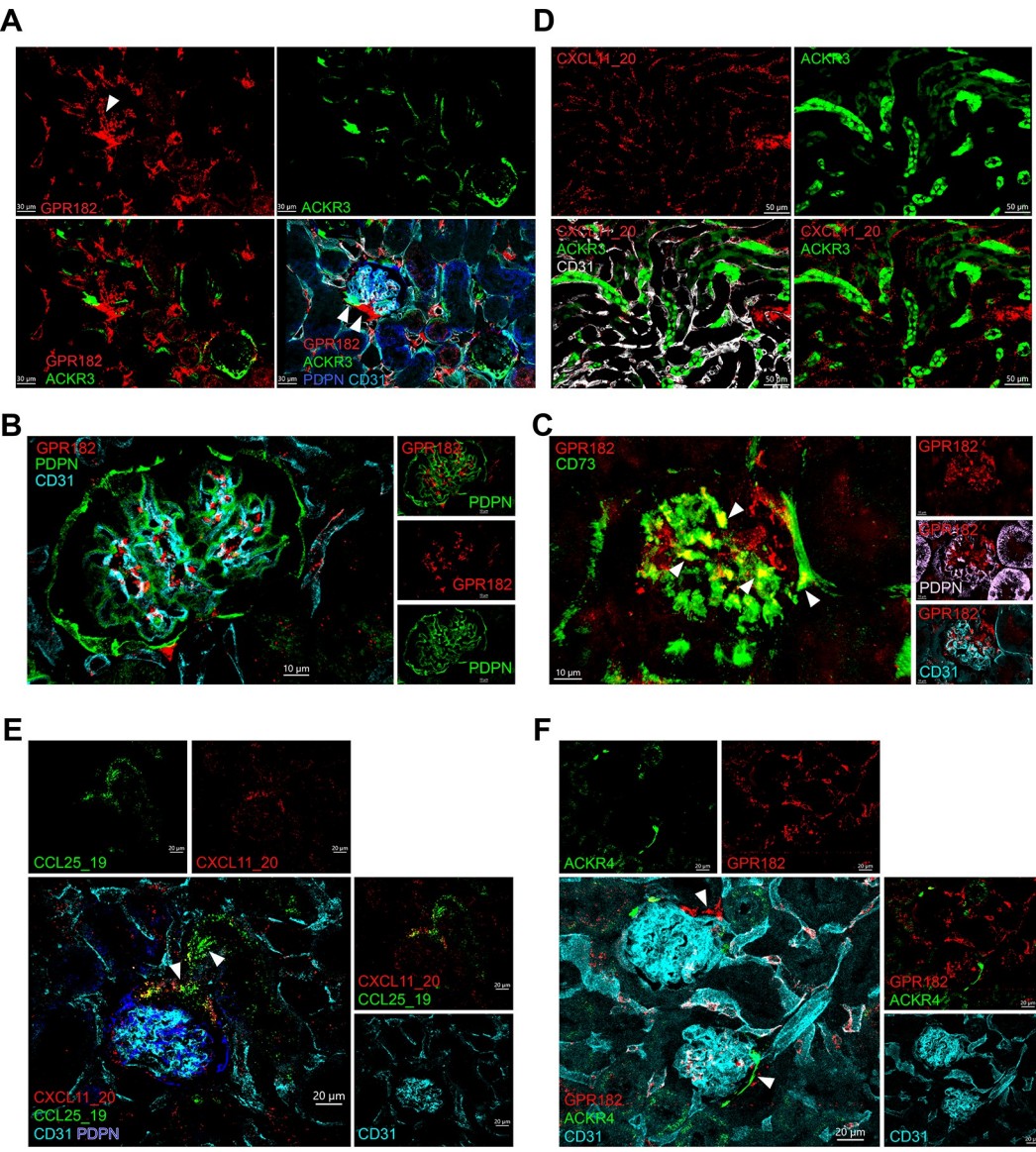

**Fig 8. ACKR3, ACKR4, and GPR182 are expressed in the kidney.** (**A**) Confocal images of kidney sections showing ACKR3 (GFP, green) and GPR182 (mCherry, red) counterstained with CD31 (cyan) and PDPN (blue) from an ACKR3^GFP/+ – GPR182^mCherry/+ mouse; arrows point to vascular pole (scale bar = 30 μm). (**B**) Images showing GPR182 (mCherry, red) expression in the glomerulus does not colocalize with CD31 (cyan) and PDPN (green) (scale bar = 10 μm). (**C**) Confocal images of GPR182 (m Cherry, red) colocalization with CD73 (green) but not with PDPN (pink) or CD31 (cyan). Arrows point to areas of colocalization (scale bar = 10 μm). (**D**) CXCL11_20 (red) was injected IV in an ACKR3^GFP/+ (GFP, green) mouse and tubuli of the kidney imaged (scale bar = 50 μm). (**E**) Coinjection of CXCL11_20 (red) and CCL25_19 (green) in a wild-type mouse highlights the glomerular vascular pole, CD31 (cyan), and PDPN (blue), scale bar = 20 μm. (**F**) The ACKR4^GFP/+ – GPR182^mCherry/+ double reporter confirms ACKR4 (GFP, green) and GPR182 (mCherry, red) coexpression in the vascular pole (scale bar = 20 μm). GFP, green fluorescent protein; PDPN, podoplanin.

antibodies, we took 2 approaches: We generated reporter mice and engineered selective chimeric chemokines. The fluorescent reporter mice that we used all contain one of the alleles for either ACKR3, ACKR4, and/or GPR182 substituted with GFP or mCherry.

In the spleen, one of the most highly perfused organs, we found elevated expression of all 3 scavengers in sinusoids and larger veins, both in common and unique locations. ACKR4, shown to be important for T cell homing [45], is strategically situated in vessels demarcating

the perimarginal sinus. The function of ACKR3 in splenic venous vasculature was suggested to be clearing the blood of excess circulating chemokines reducing the ability of chemokines to retain leukocytes (30;31). The function of GPR182 in this organ is currently unknown but could be similar to that of ACKR3. As shown for ACKR3, the highly branched sinusoids connecting into larger vessels indicate a large network of blood collecting venous system. All 3 receptors were also identified in liver sinusoids and in the vascular pole of the kidney. The high perfusion of these organs and their prominent role in blood filtration might suggest a role in depleting the bloodstream of circulating chemokines, thereby regulating their levels. However, this is yet to be confirmed.

In the LN, ACKR4 has been previously shown to play an important role in the SCS generating CCL21 gradients for DC migration into the parenchyma. We also found GPR182 to be expressed in the SCS, although less prominently and less specifically (found in both cLECs and fLECs). Coupled to the probable lower affinity of GPR182 for CCL21 (compared with ACKR4), we speculate that GPR182 might only have a secondary role in this location and that ACKR4 remains the most important scavenger in the SCS [51]. However, GPR182 was found highly expressed in HEVs, but not in the remaining thinner CD31[+] vessels that branch out of HEVs, suggesting a localized role for the receptor in HEVs, a site of leukocyte entry to LN parenchyma. Of note, ACKR1 is also known to be expressed in HEVs [72]. The role and interplay between these receptors in the LN remains to be elucidated. ACKR2 was shown to be expressed on lymphatic vessels where it efficiently scavenged inflammatory chemokines, thus preventing the activation and retention of dendritic cells within the lymphatics and the SCS of LNs and providing for optimal adaptive immune responses [73].

GPR182 was the most prominent scavenger detected in the vasculature of generative lymphoid organs. ACKR3 was found in very few lymphatic vessels in the thymus, while GPR182 was detected very broadly in the CD31[+] Lyve-1–negative vessels. Conversely, ACKR4 is detected in thymic cortical epithelium. In BM, GPR182 was also identified in the sinusoids, while ACKR3 was found in bone osteocytes. Le Mercier and colleagues suggested that expression of GPR182 here is required to scavenge CXCL12, thereby retaining hematopoietic stem cells [51]. This conclusion appears paradoxical, considering that the inhibition of CXCL12-mediated activation of CXCR4 with AMD3100 leads to stem cell mobilization [74–76]. However, high levels of endogenous CXCL12 were shown to down-regulate CXCR4 and to prevent cell migration [26]. Interestingly, ACKR4 was found in CARs. A recent transcriptomic analysis of BM niches revealed that these are predominantly adipo-CAR and osteo-CAR, which are also professional cytokine-producing cells that may be important for establishment of perivascular micro-niches [71]. Understanding the purpose of this differential positioning in the primary lymphoid organs is an interesting open question for future endeavors.

Analysis of expression in the intestine revealed expression of GPR182 in the central lymphatic, while ACKR3 was detected in the CD31[+] Lyve-1[−] blood capillaries. Both were found in submucosal vessels and ACKR4 in stromal cells. Similarly to Thomson and colleagues [77], we detected ACKR4 expression in cells of mesenchymal origin that were in contact with interstitial Cajal cells [78]. The promiscuity of ACKR chemokine ligands, which bind canonical receptors too, renders specific in vivo targeting of ACKRs challenging. The 3 selective fluorescent chemokine chimeras allowed us to confirm data from the reporter mice and detect cell surface expression and scavenging. The disadvantage of GFP reporters is the tissue intrinsic autofluorescence due to the presence of chemical compounds and metabolites that naturally emit fluorescence. Furthermore, as the receptors themselves are untagged, the fluorescent signals inevitably appear cytoplasmic making impossible to visualize precise subcellular localization. This could be partly overcome by injecting fluorescently labeled chemokines to visualize in vivo uptake. For instance, intravenous injection and subsequent uptake by endothelial cells

suggests binding and scavenging from the vessel luminal side, as observed in multiple organs. Moreover, reporter mice revealed expression but not activity. Injection of fluorescently labelled chemokines or chimeric chemokines can stimulate receptor internalization. As we clearly observed in the liver, chimeric chemokine coinjection revealed previously unknown graded patterns of expression, which could prompt further studies to unveil the relationship between ACKR4 and GPR182 in the liver.

The activity, function, and importance of ACKRs as scavenger receptors defining the fate and activity of chemokines in defined tissue microenvironments is rapidly emerging. With this study, we have provided a broad analysis of active expression of the chemokine scavengers ACKR3, ACKR4, and GPR182. We have also extended our toolbox by generating a chimeric chemokine that specifically binds and is taken up by GPR182, the newly identified chemokine scavenger, and confirmed the selectivity of CCL25_19 for ACKR4. We believe that this in-depth analysis of expression and coexpression will prompt further studies that will help unveil the function and intertwined relationships of chemokine scavenger receptors.

## Supporting information

**S1 Methods. Supporting methods.**
(DOCX)

**S1 Fig. GPR182^mCherry mouse generation.** Schematic presentation showing the gene targeting strategy via CRISPR/Cas9 to insert mCherry into the gpr182 locus on chromosome 10. Triangles show sites of probe insertion for genotyping PCR.
(TIF)

**S2 Fig. GPR182 and ACKR3 expression in the spleen.** (**A**) Confocal images showing GPR182 (mCherry, red) colocalization with CD31$^+$ vessels (green) in a cleared spleen (ACT-ECT) (scale bar = 20 μm). (**B**) Flow cytometry analysis on sorted GPR182-positive spleen endothelial cells stained for ICAM2, Meca32, and CD31. (**C**) ACKR3-expressing sinusoids (GFP, green) surround CD19$^+$ B cell follicles (RFP, red; [29]). Yellow B cells (colocalization of red (RFP, CD19) and green (GFP, ACKR3)) are visible in the MZ (scale bar = 50 μm). FCS files and gating strategies are available in FlowRepository (S2B Fig). GFP, green fluorescent protein; MZ, marginal zone.
(TIF)

**S3 Fig. Analysis of ACKR3, ACKR4, and GPR182 expression in CD45$^+$ cells.** (**A**) Representative flow cytometry plots showing ACKR3-GFP expression in CD45$^+$ cells, which were CD19$^+$ in the spleen (two left upper panels), and in the LN (two right upper panels). Gating on CD19$^+$ B220$^+$, and CD21$^{Hi}$ CD23$^{Low}$ identifies MZB, of which 50% are ACKR3$^+$ (lower panels). (**B**) Analysis of ACKR4-GFP expression in CD45$^+$ cells (upper panels) identifies few cells in the spleen, which are CD19$^+$, but almost none in the LN (right). Gating on CD45$^+$ B220$^+$, Fas$^+$ GL7$^+$ identifies GC B cells, of which 10% are ACKR4$^+$ (lower panels). (**C**) GPR182-mCherry expression is absent in CD45$^+$ cells in spleen (left) and LN (right). GFP and mCherry gates were set using a wild-type C57BL/6 mouse as negative control. FCS files and gating strategies are available in FlowRepository (S3A-S3C Fig). GC, germinal center; GFP, green fluorescent protein; LN, lymph node.
(TIF)

**S4 Fig. Analysis of ACKR3, ACKR4, and GPR182 expression in LN stromal cells.** (**A**) Representative flow cytometry plots of CD45$^-$ cells, separated using negative selection on digested tissue, show the presence of few ACKR3$^+$ CD45$^-$ cells, which are predominantly blood

endothelial cells (right, CD31$^+$ PDPN$^-$). (**B**) Analysis of ACKR4-GFP expression in the CD45$^-$ fraction identifies expression on LECs (right, CD31$^+$ PDPN$^+$). (**C**) Analysis of GPR182-mCherry expression in the CD45$^-$ fraction identifies expression primarily on blood (right, CD31$^+$ PDPN$^-$) and lymphatic (CD31$^+$ PDPN$^+$) endothelial cells. GFP and mCherry gates were set using a wild-type C57BL/6 mouse as negative control. FCS files and gating strategies are available in FlowRepository (S4A-S4C Fig). GFP, green fluorescent protein; LEC, lymphatic endothelial cell; LN, lymph node.
(TIF)

**S5 Fig. Enlargement of intestine and colon sections.** Magnified images of intestine from (**A**) ACKR3$^{GFP/+}$ – GPR182$^{mCherry/+}$ mouse, GPR182 (mCherry, red), ACKR3 (GFP, green), CD31 (pink), and Lyve-1 (cyan) (scale bar = 50 μm; arrow points to central lymphatic vessel), (**B**) ACKR4$^{GFP/+}$ – GPR182$^{mCherry/+}$ mouse, GPR182 (mCherry, red), ACKR4 (GFP, green), CD31 (pink), and Lyve-1 (cyan) (scale bar = 40 μm). (**C**) Magnified images of colon from ACKR3$^{GFP/+}$ – GPR182$^{mCherry/+}$ mouse, GPR182 (mCherry, red), ACKR3 (GFP, green), CD31 (pink), and Lyve-1 (cyan) (scale bar = 50 μm), and (**D**) ACKR4$^{GFP/+}$ – GPR182$^{mCherry/+}$ mouse, GPR182 (mCherry, red), ACKR4 (GFP, green), CD31 (pink), and Lyve-1 (cyan) (scale bar = 40 μm). (**E**) Immunofluorescence image of a section from small intestine (left image, with corresponding magnifications, including single channels) and colon (right image) from an ACKR4$^{GFP/GFP}$ mouse, showing ACKR4-expressing cells (GFP, green) and vimentin-positive cells (red), cell nuclei DAPI (cyan), and αSMA (blue) (scale bar = 20 μm). GFP, green fluorescent protein.
(TIF)

**S6 Fig. Chimeric hCXCL11_mCCL20 structure.** In silico structure of CXCL11_20, containing hCXCL11 (yellow) at the N-terminus and the body of mCCL20 (cyan). A ybbR13 tag is present at the C-terminus to allow for site specific labeling with phosphopantetheinyl transferase and a fluorescent labeled Co-enzyme A.
(TIF)

**S7 Fig. Expression of ACKR4 in the liver.** (**A**) Confocal images of ACKR4$^{GFP/+}$ mouse liver sinusoids (CD31, red), without amplification of the eGFP signal. Scale bar = 20 μm. (**B**) Confocal images of ACKR4$^{GFP/+}$ mouse liver sinusoids (CD31, red), showing expression of ACKR4 (anti-eGFP, green) at the segments proximal to the central vein, and gradually diminishing towards the periphery of the lobule. Scale bar = 20 μm. meGFP, enhanced green fluorescent protein.
(TIF)

## Author Contributions

**Conceptualization:** Serena Melgrati, Marcus Thelen.

**Data curation:** Serena Melgrati, Egle Radice, Rafet Ameti, Elin Hub, Sylvia Thelen, David Jarrossay, Antal Rot, Marcus Thelen.

**Formal analysis:** Egle Radice, Rafet Ameti, Elin Hub, Sylvia Thelen, Antal Rot.

**Funding acquisition:** Marcus Thelen.

**Investigation:** Serena Melgrati, Egle Radice, Rafet Ameti, Elin Hub, Sylvia Thelen, Pawel Pelczar, David Jarrossay, Antal Rot.

**Methodology:** Serena Melgrati, Pawel Pelczar.

**Resources:** Pawel Pelczar.

**Supervision:** Marcus Thelen.

**Validation:** Serena Melgrati, Marcus Thelen.

**Visualization:** David Jarrossay, Marcus Thelen.

**Writing – original draft:** Serena Melgrati, Marcus Thelen.

**Writing – review & editing:** David Jarrossay, Antal Rot.

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
