## [Editor Report · Decision Letter 0]

1 Dec 2022

Dear Dr Thelen, 

Thank you for submitting your manuscript entitled "Expression atlas of the anatomical localization of atypical chemokine receptors" for consideration as a Research Article by PLOS Biology.

Your manuscript has now been evaluated by the PLOS Biology editorial staff, as well as by an academic editor with relevant expertise, and I am writing to let you know that we would like to send your submission out for external peer review.

IMPORTANT: After discussions with the rest of the editorial team, we think that your manuscript would be a better fit as a Resource Article. We ask that you please change the article type to a 'Methods and Resources' Article upon re-submission (see details below):

Before we can send your manuscript to reviewers, we need you to complete your submission by providing the metadata that is required for full assessment. To this end, please login to Editorial Manager where you will find the paper in the 'Submissions Needing Revisions' folder on your homepage. Please click 'Revise Submission' from the Action Links and complete all additional questions in the submission questionnaire.

Once your full submission is complete, your paper will undergo a series of checks in preparation for peer review. After your manuscript has passed the checks it will be sent out for review. To provide the metadata for your submission, please Login to Editorial Manager (https://www.editorialmanager.com/pbiology) within two working days, i.e. by Dec 03 2022 11:59PM.

Kind regards,

Richard

Richard Hodge, PhD

Associate Editor, PLOS Biology

rhodge@plos.org

PLOS

---

## [Decision Letter · Decision Letter 1]

20 Jan 2023

Dear Dr Thelen,

Thank you for your patience while your manuscript "Expression atlas of the anatomical localization of atypical chemokine receptors" was peer-reviewed at PLOS Biology. Please accept my apologies for the delays that you have experienced during the peer review process. It has now been evaluated by the PLOS Biology editors, an Academic Editor with relevant expertise, and by three independent reviewers. 

The reviews are attached below. You will see that Reviewers #2 and #3 are positive about the manuscript and find your ACKR expression atlas to be well-done and useful for the field. However, Reviewer #1 raises concerns with the overall strength of the advance offered by the Resource, as well as the depth of the characterization for each organ. 

After discussions with the academic editor, we will not make the experiments requested by Reviewer #1 (e.g. knockout validation, binding profiles of the chimeric chemokines) a requirement for publication. We will not be able to accept the current version of the manuscript, but we would welcome re-submission of a much-revised version that takes into account the comments of Reviewer #2 and #3. We cannot make any decision about publication until we have seen the revised manuscript and your response to the reviewers' comments. Your revised manuscript is also likely to be sent for further evaluation by the reviewers.

**IMPORTANT - SUBMITTING YOUR REVISION**

*Re-submission Checklist*

*Published Peer Review*

*PLOS Data Policy*

*Blot and Gel Data Policy*

Sincerely,

Richard

Richard Hodge, PhD

Associate Editor, PLOS Biology

rhodge@plos.org

REVIEWS:

Reviewer #1: The manuscript "Expression atlas of the anatomical localization of atypical chemokine receptors" by Melgrati et al. was submitted as a Methods and Resources article to PLOS Biology. It uses mice expressing fluorescent reporter proteins from genes of atypical chemokine receptors and intravenous injection of semi-specific chemokine chimeras to map the anatomical localization of atypical chemokine receptors in immune organs and some other tissues.

Criteria:

According to the PLOS Biology criteria, Methods and Resources articles need to report a novel method or improvements to current methodologies that significantly outperform the state-of-the-art methods or that show the potential to address, for the first time, a pressing biological question. Ideally, these Methods should be of broad interest. Furthermore, Resources are required to be truly exceptional to spur future research.

Criticism:

The use of fluorescent reporter mice to map the anatomical localization of the expression of a gene of interest is a standard technique and, as such, fails to meet the criteria for Methods and Resources articles. Furthermore, the anatomical localization of atypical chemokine receptors in immune organs and other organs has been reported before based on fluorescent reporter mice and other techniques. This leaves intravenous injection of semi-specific chemokine chimeras to reveal scavenging activity of chemokine receptors. However, this technique has several shortcomings. The i.v.-injected chemokine has access only to receptors that face the circulation. Polarized cells in which the receptor of interest is targeted to a compartment that is not in contact with the circulation may not be labeled. The same is true for cells that do not contact the circulation. Chemokines are notorious for binding multiple chemokine receptors (conventional and atypical chemokine receptors). Selectivity of a chimeric chemokine needs to be demonstrated by assessing its binding profile versus atypical and conventional chemokine receptors. This is important, because conventional chemokine receptors may bind the chimera and contribute to the labeling pattern in vivo. Interpretation of data obtained with fluorescent chemokine chimeras is further complicated by receptor-independent binding such as binding to extracellular matrix. 

The information provided in this report remains fragmentary. This is because the paper aims to map multiple receptors in multiple organs, which precludes in depth analysis. An exceptional resource would provide detailed information for each reporter for all relevant cell types in an organ of interest. This may be complemented by using fluorescent chemokines (standard and chimeric) in wildtype and receptor knockout mice. Knockout validation is necessary for each receptor to determine whether chemokine binding depends on the receptor of interest. In this context, quantitative analysis of the signal in wildtype and knockout tissues is strictly required. Results need to be presented in a tabular summary.

In sum: 

The manuscript does not meet the strict criteria of a PLOS Biology Methods and Resources article.

Reviewer #2: This extraordinary paper develops novel atypical chemokine receptor selective probes for elegant in vivo imaging to define the broad distribution of 3 receptors at the cellular level in a broad array of immune and other organs. The paper's scope aligns well with the Methods and Resources section of PLoS Biology. A problem in understanding the in vivo function of these receptors is the lack of suitable antibodies and receptor-selective chemokines. The authors have engineered highly selective chimeric chemokines to solve the latter problem. Moreover, this approach overcomes tissue autofluorescence attendant with GFP localization in some organs and GFP provides only cytosolic localization information. The group has worked on this approach in previous papers, but the current work is clearly the most advanced and expanded particularly with regard to the new ACKR GPR182. The methods are solid and the results are presented in great detail and described with utmost attention to clarity. Nevertheless, I do have some suggestions and questions to guide what I think are necessary revisions. 

1. Title: Needs sharpening. 'Expression' and 'anatomical localization' are somewhat redundant, conditions are homeostatic, not inflammatory, and species is mouse. ACKR should be switched to Ackr throughout the paper. 

2. Abstract: indicate the study was restricted to homeostatic conditions in young mice. Inflammatory conditions might yield different results. Line 40: Why were Ackr1 and Ackr2 omitted from the study (except for Figure 2 ex vivo)? Using the word comprehensive is awkward in this context.

3. Introduction: the description of the receptors is a bit confusing. The authors make the distinction that ACKR1 is a sink but not a scavenger unlike ACKR2 which is a scavenger, delivering chemokines for lysosomal degradation. Then refer to ACKR3 as a scavenger and a sink without describing how it might be both. They should stick to biochemical functions or else define these metaphorical terms more exactly so there is no confusion. Also there is no introduction to the beta arrestin signaling capacity of ACKR3 and its potential importance. The authors should indicate to what extent the CXCL12 binding function of ACKR3 has been shown to explain the cardiac valve stenosis seen in ACKR3 ko mice, and the evidence that it plays a role in HSC retention in BM niches. 

4. Line 108: the paper in Molecular Cell on the mechanism of ACKR3 control of cardiac valve development also suggested an interaction with adrenomedullin.

5. Line 117: ACKR5 implies that it does not activate a G protein. To what extent has that been investigated?

6. Line 317: this implies that there is such a thing as an arteriolar sinusoid.

7. Line 363: what are the ACKR4+ 'non-endothelial Lyve-1 and CD31 negative cells of the submucosa'?

8. Line 501: what cells in the bone are expressing ACKR3?

9. Line 502: why would scavenging CXCL12 help to retain HSCs in the niches, when CXCL12-CXCR4 signaling is known to be critical for retention?

10. The Discussion should integrate the findings with what is known about ACKR2, as the authors have done for ACKR1.

11. I can't read the headings or the x-axis labels in figure 2D. Too small.

Reviewer #3: This methods and resources paper by Melgrati and colleagues describes the generation of a reporter mouse for the most recently characterised atypical chemokine receptor GPR182. These have been analysed in detail along with previously described reporters of the atypical chemokine receptors ACKR3 and ACKR4. In addition, a novel tagged chimeric chemokine (CXCL11_20) has been generated that appears to function as a specific target of GPR182. This has been used to trace functional GPR182 expression in vivo. Overall this paper adds to the reagent base available to study atypical chemokine receptors and provides the first clear picture of the expression profile of GPR182 in vivo. The expression of ACKR3 and ACKR4 described within are generally confirmatory of previous reports with some extension of the previous analysis. In general the experiments have been performed robustly and the imaging analysis presented is very impressive.

The following comments and questions should be addressed:

1) Figure 1: It would be of interest to examine ACKR4 and GPR182 sinusoids in the spleen using tissue clearing as has been done in ACKR3 mice in 1C.

2) In Line 289 it is stated that the arrows in Fig2B right point to ACKR3 expression in large sinusoids. However, the arrows in this figure point to GPR182+ACKR3- cells? Please clarify.

3) The visualisation of expression of ACKR3 in MZ B cells that is claimed is difficult to discern from the images in Fig1B. Can a closer up image of this be provided? It would also be of interest to look more closely at ACKR4 expression in the B cell compartment given previous reports. Ideally a FACS analysis of this should be done in the reporters too. Is GPR182 also expressed in B cell lineages?

4) A FACS analysis of ACKR3 and ACKR4 in LN stroma should be presented (perhaps supplementary) to complement the FACS analysis of these cells with respect to GPR182.

5) ACKR4 has previously been reported to be expressed in fibroblasts (Thomson et al. JI 2018; Bastow et al. PNAS. 2021). Whether the ACKR4+ cells in intestine (Fig6) co-express fibroblast markers should be tested.

6) Fig 7A: Does the difference in MFI of CXCL11_20 increase in GPR182 cells signifcantly increase from 4C to 37C? If not then this would indicate that GPR182 is mediating chemokine binding/sequestration rather than uptake. Please clarify.

7) Similarly, in 7E, how is CXCL11_20 uptake distinguished from binding in the spleen endothelial cells examined here?

7) References 49 and 31 are duplications.

8) The methods states that sections were stained with anti-CD31-AF488 (line 195). It is not clear how this is spectrally distinguished from the GFP signal from the reporters that express GFP. Please clarify.

9) Line 72-73: ACKR2 is also able to scavenge CXCL10 so is not restricted to CC chemokines as implied here.

10) Figure 7F should be referred to in the text of the paragraph starting on line 413

---

## [Editor Report · Decision Letter 2]

29 Mar 2023

Dear Dr Thelen,

Thank you for your patience while we considered your revised manuscript "Atlas of the anatomical localization of atypical chemokine receptors in healthy mice" for publication as a Methods and Resources Article at PLOS Biology. This revised version of your manuscript has been evaluated by the PLOS Biology editors and the Academic Editor.

Based on our Academic Editor's assessment of your revision, I am happy to say that we are likely to accept this manuscript for publication, provided you satisfactorily address the following data and other policy-related requests (A-E):

(A) In the Methods section of the manuscript, please provide the specific name of the Institutional Animal Care and Use Committee (IACUC) that reviewed and approved the study (i.e. name of affiliated institution). In addition, please provide the specific approval number issued by the IACUC to conduct the study. 

(B) You may be aware of the PLOS Data Policy, which requires that all data be made available without restriction: http://journals.plos.org/plosbiology/s/data-availability. For more information, please also see this editorial: http://dx.doi.org/10.1371/journal.pbio.1001797

- Supplementary files (e.g., excel). Please ensure that all data files are uploaded as 'Supporting Information' and are invariably referred to (in the manuscript, figure legends, and the Description field when uploading your files) using the following format verbatim: S1 Data, S2 Data, etc. Multiple panels of a single or even several figures can be included as multiple sheets in one excel file that is saved using exactly the following convention: S1_Data.xlsx (using an underscore).

-Deposition in a publicly available repository. Please also provide the accession code or a reviewer link so that we may view your data before publication.

Figure 2C-D, 2F, 7A, 7F

(C) For figures containing FACS data (Figure 2A, 3C, 5F, 7E, S2B, S3A-C, S4A-C), please provide the FCS files and a picture showing the successive plots and gates that were applied to the FCS files to generate the figure. We ask that you please deposit this data in the FlowRepository (https://flowrepository.org/) and provide the accession number/URL of the deposition in the Data Availability Statement in the online submission form.

(D) Please also ensure that each of the relevant figure legends in your manuscript include information on *WHERE THE UNDERLYING DATA CAN BE FOUND*, and ensure your supplemental data file/s has a legend.

(E) Please ensure that your Data Statement in the submission system accurately describes where your data can be found and is in final format, as it will be published as written there.

We expect to receive your revised manuscript within two weeks. 

*Published Peer Review History*

*Press*

Kind regards,

Richard

Richard Hodge, PhD

Associate Editor, PLOS Biology

rhodge@plos.org

PLOS

---

## [Editor Report · Decision Letter 3]

5 Apr 2023

Dear Marcus,

On behalf of my colleagues and the Academic Editor, Hans-Uwe Simon, I am pleased to say that we can accept your manuscript for publication, provided you address any remaining formatting and reporting issues. These will be detailed in an email you should receive within 2-3 business days from our colleagues in the journal operations team; no action is required from you until then. Please note that we will not be able to formally accept your manuscript and schedule it for publication until you have completed any requested changes.

PRESS

Best wishes, 

Richard

Richard Hodge, PhD

Associate Editor, PLOS Biology

rhodge@plos.org

PLOS
